# Mollifier Layers: Enabling Efficient High-Order Derivatives in Inverse PDE Learning

**Vinayak Vinayak**[*]
*University of Pennsylvania*
vinayakv@seas.upenn.edu

**Ananyae K Bhartari**[*]
*University of Pennsylvania*
ananyae@seas.upenn.edu

**Vivek B Shenoy**
*University of Pennsylvania*
vshenoy@seas.upenn.edu

**Reviewed on OpenReview:** *https://openreview.net/forum?id=6mFVZSzyev*

## Abstract

Parameter estimation in inverse problems involving partial differential equations (PDEs) underpins modeling across scientific disciplines, especially when parameters vary in space or time. Physics-informed Machine Learning (PhiML) integrates PDE constraints into deep learning, but prevailing approaches depend on recursive automatic differentiation (autodiff), which produces inaccurate high-order derivatives, inflates memory usage, and underperforms in noisy settings. We propose Mollifier Layers, a lightweight, architecture-agnostic module that replaces autodiff with convolutional operations using analytically defined mollifiers. This reframing of derivative computation as smoothing integration enables efficient, noise-robust estimation of high-order derivatives directly from network outputs. Mollifier Layers attach at the output layer and require no architectural modifications. We compare them with three distinct architectures and benchmark performance across first-, second-, and fourth-order PDEs—including Langevin dynamics, heat diffusion, and reaction-diffusion systems—observing significant improvements in memory efficiency, training time and accuracy for parameter recovery across tasks. To demonstrate practical relevance, we apply Mollifier Layers to infer spatially varying epigenetic reaction rates from super-resolution chromatin imaging data—a real-world inverse problem with biomedical significance. Our results establish Mollifier Layers as an efficient and scalable tool for physics-constrained learning.

## 1 Introduction

Parameter estimation through inverse problems constrained by PDEs arise across science and engineering, where parameters like diffusivity or conductance must be inferred from sparse, noisy measurements (Kaipio & Somersalo (2006); Tarantola (2005); Herrera et al. (2022); Zhang et al. (2020)). Traditional inference methods are computationally intensive, as they require repeatedly solving forward PDEs. These challenges grow worse in high-resolution or real-time settings, especially when governing parameters vary at fine scales or data is sparse (Xun et al. (2013)).

Deep learning has emerged as a powerful framework for such problems, with physics-informed machine learning (PhiML) approaches (Toscano et al. (2024))—like PINNs (Raissi et al. (2019)), DeepFNO (Li et al. (2020)), and DeepONet (Lu et al. (2021))—embedding PDE constraints into model training and inference. These methods enable learning from limited observations, but rely heavily on recursive automatic differentiation (autodiff) to compute spatial and temporal derivatives (Baydin et al. (2018)). Autodiff based

---

[*]These authors contributed equally.

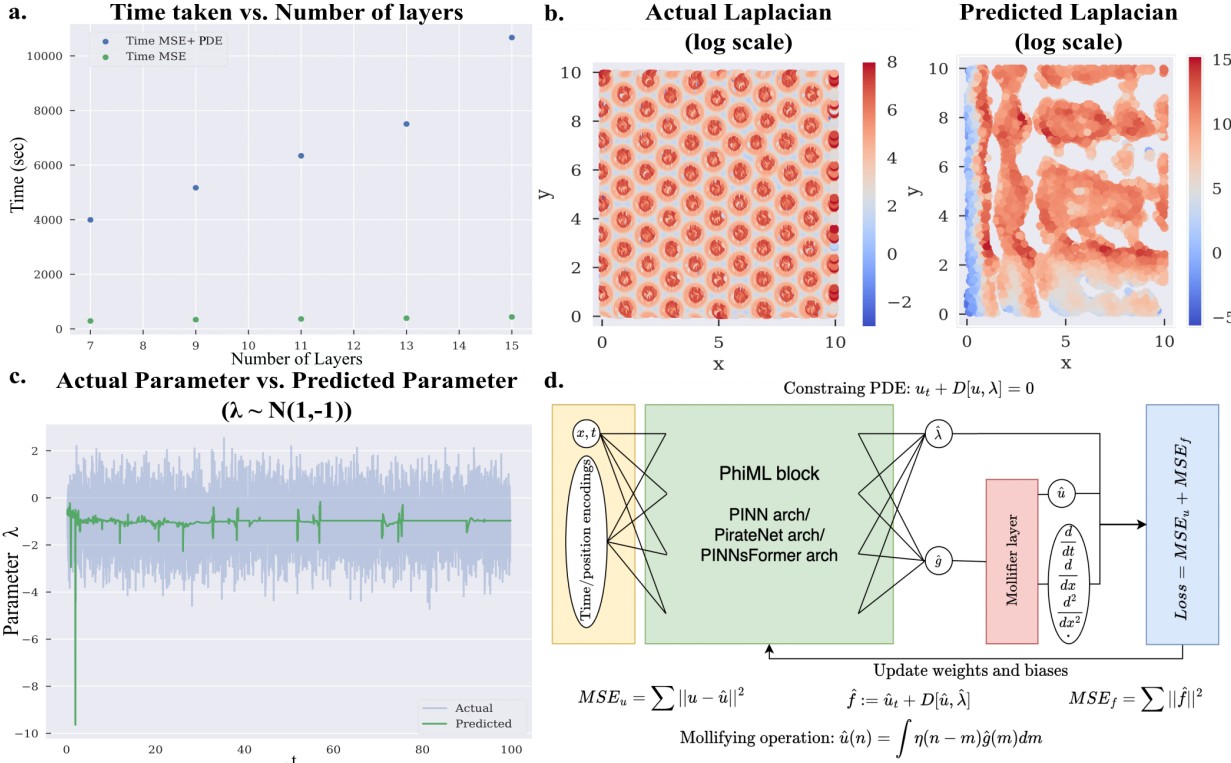

Figure 1: Limitations of autodiff and overview of PhiML+Mollifier architecture. **(a)** Training time comparison for PINNs with and without PDE residual loss. **(b)** PINN-predicted vs. actual Laplacian in a reaction-diffusion system. **(c)** PINN-predicted vs. actual forcing term in a Langevin system. **(d)** PhiML+Mollifier Layer architecture replacing autodiff with mollifier-based convolution.

PhiML techniques are memory-intensive and can be unstable for higher-order derivatives (Wang et al. (2024); Fuks & Tchelepi (2020); Fourment et al. (2023); Margossian (2019)), making such models brittle under noise and costly for deep architectures.

To overcome these limitations, we propose *mollifier layers* (inspired from Friedrichs (1944); Murio et al. (1998)): lightweight, architecture-agnostic modules that replace autodiff with convolution-based derivative estimation using smooth, compactly supported kernels. These layers estimate derivatives via convolution with analytic derivatives of smooth mollifier kernels, transferring differentiation off the network and implicitly regularizing the computation for stability and noise robustness.

Integrated into and benchmarked against three PhiML architectures—PINNs (Raissi et al. (2019)), PirateNet (Wang et al. (2024)), and PINNsFormer (Zhao et al. (2023))—mollifier layers offer a scalable, robust alternative to autodiff for physics-informed learning. They achieve significant improvements in parameter recovery, derivative stability, and computational efficiency across diverse PDEs, including a fourth-order reaction-diffusion model (Heo et al. (2023)) where they infer the governing parameter with high correlations while reducing the memory usage and time complexity by over an order of magnitude. Applied to super-resolution microscopy (Schermelleh et al. (2019)) of the human cell nuclei, they enable spatially resolved inference of critical biophysical parameters from noisy data, demonstrating practical utility in real-world scientific modeling.

While our benchmarks target inverse PDEs, the underlying principle—offloading high-order derivative computation to analytic mollifier convolutions—can possibly generalize to forward models (Karniadakis et al. (2021)) (shown briefly in B.5), operator learning (Azizzadenesheli et al. (2024)), or neural ODE systems (Ruiz-Balet & Zuazua (2023); Chen et al. (2018)) where gradient accuracy is crucial.

Table 1: Peak memory usage by PINNs

| Equation | MSE Loss Only | MSE + PDE Loss |
|---|---|---|
| Langevin (first order 1D) | 0.17 GB | 0.21 GB |
| Heat (second order 2D) | 0.80 GB | 1.20 GB |
| Reaction-Diffusion (fourth order 2D) | 0.40 GB | 2.70 GB |

## 2 Motivation

As shown in prior work (Wang et al. (2024); Fuks & Tchelepi (2020); Fourment et al. (2023); Margossian (2019)), recursive autodiff—despite its central role in PhiML—poses key bottlenecks for learning PDEs, particularly in the accurate and efficient estimation of higher-order derivatives. To expose and quantify these, we benchmarked an *upgraded PINN* (with fourier-features and residual connections, details in A.1), which we will refer to as PINN on spatio-temporally varying parameter estimation. This analysis revealed three core issues: compute cost, derivative inaccuracy, and memory blow-up, that directly motivated our mollifier layer design.

**Motivation #1: Rising cost of high-order derivatives with model depth.** Evaluating PDE residuals in PhiML requires multiple backward passes for each order of differentiation, causing compute time to grow superlinearly with network depth (Cho et al. (2024); Suarez Cardona & Hecht (2023)). This becomes especially burdensome in complex systems that demand deeper networks. For instance, in the reaction-diffusion benchmark (system described in Sec. 4.3), training PINNs with 7 to 15 layers using the full loss (data + PDE) took significantly longer than using the data loss alone—an overhead that widens with network depth (Fig 1a). Such costs slow down training and escalate power/hardware requirements, forming a major scalability barrier.

**Motivation #2: Inaccuracy in higher-order derivatives for high-frequency features.** Recursive autodiff is susceptible to rounding errors from repeated gradient operations. This instability becomes particularly problematic when modeling spatio-temporally varying or noisy parameters. For example, a PINN trained to recover the gradient in a reaction-diffusion system achieved only 0.21 Laplacian correlation with ground truth (Fig 1b), indicating poor quantitative accuracy.

These issues are exacerbated with high-frequency or noisy signals, where the inherent smoothness of neural networks hinders the capture of sharp or discontinuous features (Fuks & Tchelepi (2020)). In a Langevin system with Gaussian white noise (Sec. 4.1.1), the PINN failed to capture variance in the inferred parameter (Fig 1c). Together, these results underscore the fragility of autodiff in noisy, high-order settings, motivating the need for more stable and noise-resilient derivative estimation techniques.

**Motivation #3: High memory overhead from recursive autodiff.** Computing higher-order derivatives through recursive autodiff requires storing intermediate gradients across layers, leading to memory usage that scales with both depth and derivative order. In experiments with parameter estimation for Langevin, heat, and reaction-diffusion equations using PINN(Sec. 4), adding the PDE error significantly increased GPU memory usage (hardware specs in A.6) compared to training with data-only losses (Table 1). This memory blow-up limits scalability and highlights the need for more memory-efficient alternatives.

## 3 Methods

### 3.1 Formulating parameter inference using PhiML

To ground our innovation, we begin with a brief overview of parameter inference for PDEs using a general PhiML framework. Without loss of generality, consider a partial differential equation:

$$u_t + D[u, \lambda] = 0, x \in \Omega, t \in T \tag{1}$$

where $u(t,x)$ is the solution of the PDE, $D[.;\lambda]$ is a non-linear/linear ordinary or partial differential operator parameterized by $\lambda(t,x)$ and the domain $\Omega$ is a subset in $R^n$. Subscripts denote differentiation in time $t$ or space $x$. Given this PDE, the aim is to infer $\lambda(t,x)$ given a sparsely sampled set of $u(t,x)$. In PhiML, the solution $u(t,x)$ is approximated by a deep neural network $u_\theta(t,x)$ (or $\hat{u}$), where $\theta$ represents the set of all trainable parameters of the network. With the autodiff formulation (Paszke et al. (2017), Raissi et al. (2019)), required gradients with respect to input variables $(t,x)$ or network parameters $\theta$ can be computed. Additionally, $\hat{\lambda}$, the estimated parameter, is also an output of the neural network. The actual parameter $\lambda(t,x)$ is spatially or temporally varying, which can be due to a patterned modulation in these parameters or fluctuations due to noise. We assume that $\lambda(t,x)$ is unknown across $\Omega$, making parameter inference challenging but broadly applicable. Combining the estimates $(\hat{u},\hat{\lambda})$ with the derivatives of $\hat{u}$ computed via autodiff, the PDE residual is defined as $\hat{f} = \hat{u}_t + D[\hat{u},\hat{\lambda}]$, evaluated at all points in $\Omega$. With these tools, the optimization problem is to minimize the total mean squared error ($MSE_{total}$), which is defined as $MSE_{total} = MSE_u + MSE_f$, where,

$$\text{Data loss: } MSE_u = \frac{1}{N_u} \sum_{i=N_u} |\hat{u}(t^i, x^i) - u(t^i, x^i)|^2 \tag{2}$$

and

$$\text{PDE residual loss: } MSE_f = \frac{1}{N_f} \sum_{j=N_f} |\hat{f}(t^j, x^j, \lambda^j)|^2. \tag{3}$$

Here, $N_u$ are the training data points available for the actual solution $u(t,x)$ and $N_f$ are all the points in the discretized domain for which the solution is being sought. For the initial testing of the models and characterizing their behavior, we use $N_u = N_f$ and later validate the models on $N_u = 0.1N_f$, i.e., with 10% sampling. The data loss is calculated on $N_u$, while the PDE loss on $N_f$.

### 3.2 Modified parameter inference with Mollifier Layers

*Intuition behind Mollifier Layers:* Recognizing the limitations of recursive autodiff, we draw from the weak-form formulation in finite element methods (Hughes (2003)), where derivatives are inferred via integration against smooth test functions. This principle suggests an alternative to gradient chaining: recovering derivatives through structured integration applied directly to the network output. A similar idea underlies Savitzky–Golay filters (Savitzky & Golay (1964)), which compute stable numerical derivatives by fitting local polynomials within a sliding window—smoothing and differentiating simultaneously to mitigate the effects of noise and discretization.

Mollifier layers formalize this intuition within neural architectures: rather than relying on recursive autodiff, they compute derivatives via convolution with analytic derivatives of smooth, compactly supported kernels. These mollifiers serve as localized test functions, converting differentiation into a stable smoothing operation that is mathematically grounded and memory-efficient. Compared to alternatives like Gaussian filters or splines, mollifiers offer two key advantages: (1) compact support for localized and efficient computation, and (2) closed-form derivatives for stable gradient flow. Prior analysis (Murio et al. (1998)) confirms that this formulation achieves provably bounded error under discretization and measurement noise, making it well-suited for high-order and noise-prone inverse problems.

*Formal definition:* Drawing inspiration from Mollifiers (Friedrichs (1944)) and the weak form formulation of PDEs (details in A.7), we construct a modified estimator, as illustrated in Fig 1d. Although the base PhiML architecture remains unchanged aside from an added mollifier layer at the output, our methodology diverges from traditional PhiML approach in key ways:

- Rather than predicting the target field $\hat{u}$ directly, the base network outputs $\hat{g}$. This decouples function approximation from derivative evaluation: all PDE derivatives needed for the residual are computed at the Mollifier Layer from $\hat{g}$ using fixed (non-learned) operators, avoiding recursive autodiff through the network.

- $\hat{u}$, is obtained at the Mollifying layer through the following operation (where $*$ is the convolution operator, $\eta$ is the Mollifying function and $U$ is the compact support of $\eta$):

$$\hat{u}(n) = (\hat{g} * \eta)(n) = \int_{m \in U} \hat{g}(m)\eta(n - m)\,dm, \tag{4}$$

- At the Mollifier Layer, the higher-order derivatives w.r.t. a variable $j$ (e.g., time or space) at a coordinate $n$, such as $\hat{u}_j(n)$, $\hat{u}_{jj}(n)$, etc., are obtained through the following operations:

$$\hat{u}_j(n) = (\hat{g} * \eta_j)(n) = \int_{m \in U} \hat{g}(m)\eta_j(n - m)\,dm, \tag{5}$$

$$\hat{u}_{jj} = (\hat{g} * \eta_{jj})(n) = \int_{m \in U} \hat{g}(m)\eta_{jj}(n - m)\,dm. \tag{6}$$

This formulation offloads derivative computation to analytic derivatives of a predefined mollifier $\eta$. Rather than using recursive autodiff, we obtain the required derivatives by convolving the learned function $\hat{g}$ with the corresponding kernel derivatives $\eta_j$, $\eta_{jj}$, etc.; any order-$k$ derivative is computed directly using the precomputed order-$k$ derivative of $\eta$. *This decoupling is the central innovation: it enables efficient and stable high-order derivatives, inherently regularized by a low-pass filtering effect, while substantially reducing memory and compute costs.*

**Implementation of Mollifier Layers.** In practice, the integrals in Eqs (4-6) are implemented as discrete convolutions over the compact support $U$. The network outputs $\hat{g}$ on the collocation grid, and the Mollifier Layer applies fixed kernels to produce $\hat{u} = \hat{g} * \eta$ and derivative fields such as $\hat{u}_j = \hat{g} * \eta_j$ and $\hat{u}_{jj} = \hat{g} * \eta_{jj}$. Consequently, each required PDE derivative is obtained by a single convolution with the corresponding analytic kernel derivative, avoiding recursive autodiff through the network. In Fig 1d, the Mollifier Layer corresponds exactly to these fixed-kernel convolutions, with kernel size determined by $U$. During training, we compute the PDE residual loss from $\hat{u}$ and its derivatives; gradients backpropagate through the convolution operations to update the base network parameters, while the mollifier kernels $(\eta, \eta_j, \eta_{jj})$ remain fixed.

The mollifier $\eta$ is designed to satisfy the following properties:

1. Smoothness (Infinitely Differentiable): The mollifier $\eta$ is chosen from the $C^\infty$ class to enable computation of all high-order PDE derivatives.

2. Compact Support: $\eta$ has compact support on $U \subset \mathbb{R}^n$, meaning $\eta(m) = 0$ for all $m \notin U$. This confines integration to a finite region, reducing computational cost by restricting convolution to a fixed kernel size.

3. Non-Negativity: $\eta(m) \geq 0$ for all $m \in U$, ensuring the mollifier behaves as a localized averaging kernel and adheres to the following:

   (a) In the limit the support of $\eta$ contracts to zero, the mollifying operation converges to the identity operator, with $\eta$ approximating the Dirac delta $\delta(m)$.
   (b) Non-negativity prevents destructive interference during integration, mitigating cancellation errors common in oscillatory kernels.
   (c) In physical systems with inherently non-negative targets (e.g., density, concentration), non-negative mollifiers preserve this constraint for consistent estimates.

We consider several mollifiers (e.g., different order polynomials and sine-based kernels) defined on appropriate compact supports; their explicit forms are provided in App A.5. An extension of mollifier layers to multi-parameter (vector-valued) inverse problems is given in App D. In our default setting we use a smooth compactly supported mollifier, e.g. the standard $C^\infty$ bump $\eta(x) = c \exp\left(-\frac{1}{1-\|x\|^2}\right) \mathbf{1}_{\{\|x\|<1\}}$. The derivative kernels $\eta_j, \eta_{jj}, \ldots$ are obtained by analytic differentiation of $\eta$ and discretized over the same support $U$ used

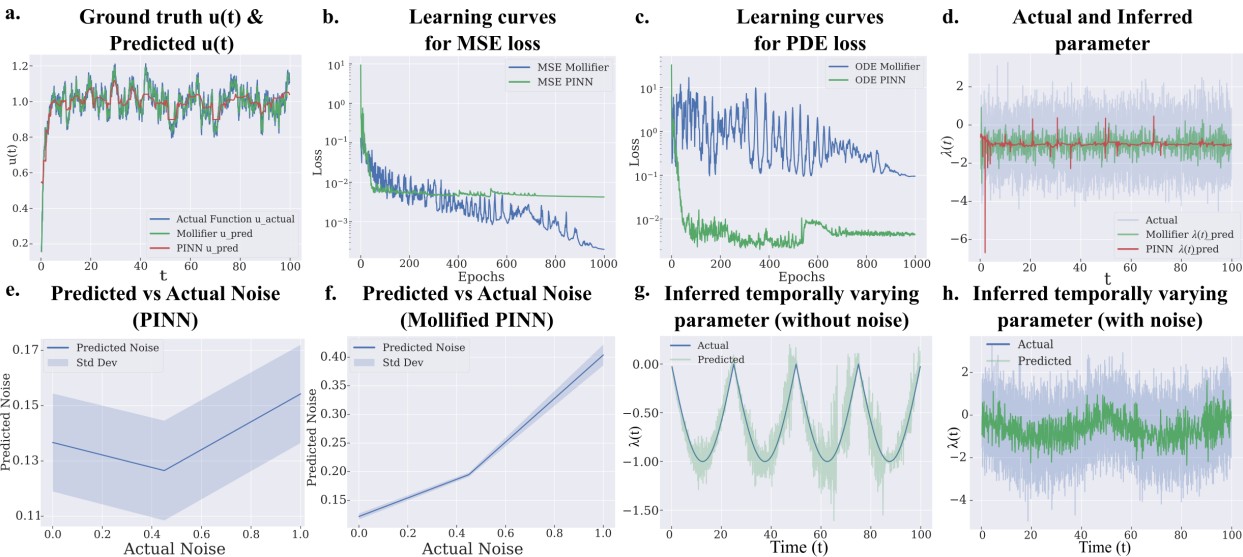

Figure 2: Parameter inference for the Langevin equation using PINN and Mollified PINN. a) Ground truth and predicted $u(t)$. b–c) $MSE_u$ Data and $MSE_f$ PDE residual learning curves. d) Ground truth and inferred forcing term $\lambda(t)$ under Gaussian noise. e–f) Actual vs. predicted noise trends in $\lambda(t)$. g) Inferred time-varying $\lambda(t)$ without noise. h) Inferred time-varying $\lambda(t)$ with added noise.

in Eqs (4-6). The kernel size is determined by the compact support $U$ and controls the smoothing detail trade-off; we sweep a small set of supports and use validation performance to select $U$ (values in Sec A.5).

With this formulation, mollifier layers provide the following key advantages over traditional autodiff-based physics-informed approaches: **(1) Computational efficiency:** Derivatives are computed via a single output-layer convolution, avoiding recursive autodiff and reducing memory usage. **(2) Structural decoupling:** Differentiation is shifted to fixed analytical kernels, independent of network depth. **(3) Noise robustness:** As localized smoothers, mollifiers suppress high-frequency artifacts, enabling stable inference under noise.

Convergence guarantees of the Mollifier layer. [Uniform derivative consistency] Let the true field $u \in C^1([0,1])$ be $L$-Lipschitz. Noisy grid samples are $g_j = u(x_j) + n_j$ at $x_j = jh$ with $|n_j| \leq \varepsilon$. Denote by $\eta_\delta(r) = \delta^{-1}\eta(r/\delta)$ the analytic kernel from the mollifier layer and $J_\delta g = \eta_\delta * g$. Then

$$\left\| D_0(J_\delta g) - u' \right\|_\infty \leq C_1\,\delta + C_2\,(h+\varepsilon), \tag{7}$$

where $D_0 g_j = (g_{j+1} - g_{j-1})/(2h)$. [Proof sketch.] Combine Murio *et al.* Murio et al. (1998) Thm 2.6 (continuous bias) with Thm 3.5 (discrete + noise); see appendix A.8. Choosing $\delta \approx \sqrt{h}$ balances bias and variance, giving $\mathcal{O}(\sqrt{h})$ uniform error while the backward operator norm remains $\mathcal{O}(1/\delta)$.

### 3.3 Inverse parameter estimation

The proposed methodology estimates the spatio-temporally varying PDE parameter $\hat{\lambda}(t,x)$. As discussed in Motivation#2, even though this approach yields estimates for smoothly varying parameters; it may underperform when capturing high-frequency variations due to the inherent smoothness bias of neural network outputs. To mitigate this limitation, we introduce an alternative estimation strategy applicable when $\lambda(t,x)$ can be expressed as a function of other predicted quantities—i.e., when $\lambda$ is separable. For example, consider a PDE of the form $u_t - \lambda D[u] = 0$, then our final estimated PDE parameter is $\hat{\lambda}_{final} = \frac{\hat{u}_t}{D[\hat{u}]}$. We propose that this estimation will better capture the ground truth, as $\hat{u}$, optimized through the data loss ($MSE_u$), reflects data variance and is better equipped to capture rapidly varying spatio-temporal features. We emphasize that this is a special case; the mollifier layer applies regardless of whether $\lambda$ is separable.

## 4 Experiments

We evaluate our method on three PDE benchmarks of increasing complexity: 1D Langevin (first-order), 2D heat (second-order), and 2D reaction–diffusion (fourth-order). We compare standard architectures—*upgraded PINNs* (Raissi et al. (2019)) (with residual connections (Wang et al. (2021a)), spatio-temporal encodings (Wang et al. (2021b)), and layer norm (Ba et al. (2016))), PirateNet (Wang et al. (2024)), and PINNsFormer (Zhao et al. (2023))—against their mollified counterparts where applicable. These pairwise comparisons (native vs. native+mollifier) serve as direct ablation studies, isolating the impact of mollifier layers across models. The selected architectures reflect a spectrum of design choices: PINNs as the canonical baseline, PirateNet for its stability with higher-order derivatives, and PINNsFormer for its use of attention mechanisms (Vaswani et al. (2017); Dosovitskiy et al. (2020); Niu et al. (2021)). Due to the high computational cost of attention-based models, we do not include a mollified PINNsFormer; mollified PINNs already achieve superior performance at a fraction of the computational cost. To further benchmark against alternatives to autodiff, we compare our method with FD-PINNs, which approximate derivatives using finite differences (Lim et al. (2022); Huang & Alkhalifah (2024); Chiu et al. (2022)). While FD-PINNs are among the most memory- and time-efficient approaches, our results show that Mollifier Layers, by virtue of their built-in regularization, achieve superior accuracy—particularly for higher-order derivatives (theoretical comparison in A.8). While all quantitative comparisons are reported in the main text, we plot only the Mollified-PINN variants for clarity. All supporting plots are provided in the appendix.

### 4.1 First-Order 1D Langevin equation

To evaluate parameter inference under time-varying dynamics, we consider a simplified form of the Langevin equation—a classical model for systems subject to both deterministic and stochastic forces. Specifically, we use the rescaled first-order ODE:

$$u_t = u + \lambda(t), \tag{8}$$

where $u(t)$ is the observed velocity and $\lambda(t)$ is an unknown forcing term to be inferred. This formulation preserves the essential Langevin-dynamics structure—modeling stochastic forcing transients—while sidestepping second-order complexity. As our simplest nontrivial ODE, it offers a clear benchmark for evaluating inference accuracy and robustness.

**Setup.** We observe $u(t)$ at $N_u$ time points and infer the forcing $\lambda(t)$ in $u_t = u + \lambda(t)$. Training minimizes a data misfit on $u(t)$ together with the PDE residual enforced at $N_f$ collocation points; derivatives are computed via autodiff for native models and via the Mollifier Layer for mollified variants. We report (i) correlation of the inferred forcing mean and variance against ground truth and (ii) training time and peak memory (Table 2; Fig. 2). Unless stated otherwise, we use the default mollifier $\eta$ from Sec. A.5. Under constant forcing $\lambda$, all models perform similarly; we therefore focus below on settings with temporal variation and/or noise where derivative stability matters most.

#### 4.1.1 Langevin equation with Gaussian White Noise

We simulate trajectories where the forcing is $\lambda(t) \sim \mathcal{N}(\Lambda, \sigma^2)$ with $\Lambda = -1$ being the mean of the forcing term and $\sigma \in \{0, 0.44, 1\}$ being the variance capturing the noise around the mean, and we provide observations of $u(t)$ to the models while treating $\lambda(t)$ as unknown. All methods optimize the standard inverse-PDE objective consisting of a data term on $u(t)$ and a physics residual term for $u_t - u - \lambda(t)$, with mollified derivatives used for the residual in mollified variants. For consistency, we set $N_u = N_f$ across all experiments, using the default mollifier $\eta$ (Sec. A.5). As shown in Fig. 2a–c for $\sigma = 1$, both PINNs and mollified PINNs converge to the underlying function, but the latter, with our parameter estimation scheme, more accurately captures noise variations (Fig. 2d). Across noise levels, mollified PINNs consistently outperform standard PINNs in capturing these trends (Fig. 2e). The same qualitative improvements hold across architectures (Table 2).

We evaluated each method's accuracy—via mean and variance correlation—and computational efficiency—via training time and memory—over five runs per noise level. Mollified variants consistently outperformed their native counterparts in tracking temporal trends and used fewer resources on average (Table 2).

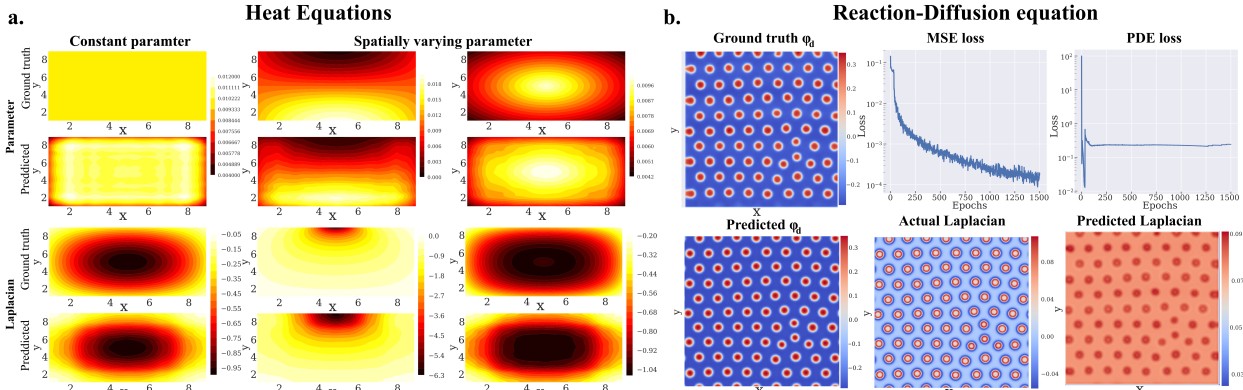

Figure 3: Parameter estimation for the Heat and Reaction–Diffusion equations using Mollified PINN. a) Ground truth and predictions for constant and spatially varying thermal diffusivity $\lambda(x, y)$ and Laplacians for the 2D heat equation. b) Predicted vs. actual chromatin order parameter $\phi_d$, recovered Laplacian, and training curves for the reaction–diffusion system.

### 4.1.2 Langevin Equation with temporally varying force

We next evaluate the performance of the models in inferring a noiseless, temporally varying parameter. For this, we simulate a trajectory with a time-dependent mean forcing term, $\lambda(t) = \Lambda(t)$, and generate simulations for various configurations of $\Lambda(t)$ to compare the capabilities of the native models with their mollified versions. As shown in Fig. 2g, mollified PINNs provide a very good estimate of the underlying parameter's variation. While all variants successfully capture the global mean, native variants generally fail to capture the temporal variation in the parameter.

We introduced noise ($\sigma = 1$) to the underlying parameters by sampling $\lambda(t) \sim \mathcal{N}(\Lambda(t), 1)$. As illustrated in Fig. 2h, mollified PINNs demonstrate great performance in capturing temporal trends in the forcing term compared to PINNs (Fig. 13). Similar trends are obtained for the other models (Figs. 14–16), proving that the mollifier layer lends higher accuracy predictions.

Additional results (constant forcing, other architectures, and kernel ablations) are in App. B (Secs. B.1–B.4; Figs. 5–16).

### 4.2 Second-Order 2D Heat Equation

We consider the steady-state 2D heat equation, a second-order PDE modeling spatial temperature distribution under thermal diffusion and external sources:

$$\lambda(x, y)\nabla^2 u + m(x, y) = 0, \tag{9}$$

where $u(x, y)$ is the observed temperature field, $\lambda(x, y)$ is the unknown thermal diffusivity, and $m(x, y)$ is a known source term. The goal is to recover $\lambda(x, y)$ from a sparsely sampled $u(x, y)$ and a known $m$, a problem relevant in materials science, geophysics, and biomedical imaging for characterizing heterogeneous media.

**Setup.** We observe $u(x, y)$ at $N_u$ spatial locations and infer $\lambda(x, y)$ in $\lambda\nabla^2 u + m = 0$, with $m(x, y)$ known. Training minimizes a data term on the observed $u$ together with the PDE residual enforced at $N_f$ collocation points, using autodiff for native models and the Mollifier Layer to compute $\nabla^2 u$ for mollified variants. We report correlation between inferred and ground-truth $\lambda(x, y)$ and computational cost (training time and peak memory; Table 2).

We evaluate all models on cases with both constant and spatially varying diffusivity. Fig 3a shows representative results for mollified PINNs, and Table 2 summarizes correlation and efficiency across methods. Additional qualitative comparisons for other architectures are provided in App. B (Figs 17-20).

### 4.3 Fourth-Order 2D Reaction–Diffusion Equation

Reaction-diffusion systems model spatial pattern formation via diffusion and local reaction kinetics, with applications in biology, chemistry, and ecology (Epstein & Xu (2016)). We study a biologically motivated system describing DNA organization in the eukaryotic/human nucleus, where heterochromatin (inactive) and euchromatin (active) states are shaped by diffusion and epigenetic reactions such as methylation and acetylation—key drivers of gene regulation (Kant et al. (2024); Vinayak et al. (2025); Dhankhar et al. (2025)). Our goal is to infer the spatially varying reaction rate $\lambda(x, y)$ from the final chromatin configuration in synthetically generated data, enabling scalable estimation of biophysical parameters. In Sec. 5, we extend this framework to experimentally obtained super-resolution imaging data.

We adopt a phase-field model (Kant et al. (2024)) with volume fractions $\phi_h$, $\phi_e$, and $\phi_n$ for heterochromatin, euchromatin, and nucleoplasm, respectively. These volume fractions satisfy the space-filling constraint $\phi_h + \phi_e + \phi_n = 1$. In Sec 4.3 we focus on the inverse problem for $\lambda(x, y)$ using the chromatin order parameter $\phi_d = \phi_h - \phi_e$; $\phi_n$ is therefore omitted from subsequent expressions (it can be recovered implicitly from the constraint when needed). The dynamics of the system are described as:

$$\frac{\partial \phi_d}{\partial t} = \nabla^2 \mu_d + 2(\lambda \phi_e - \phi_h), \tag{10}$$

where $\mu_d$ is a chemical potential depending on $\phi_h$ and $\phi_e$. At steady state, this yields:

$$\nabla^2 \mu_d + 2(\lambda \phi_e - \phi_h) = 0. \tag{11}$$

Since $\mu_d$ depends on second-order derivatives of $\phi_d$ (i.e., includes terms proportional to $\nabla^2 \phi_d$), the operator $\nabla^2 \mu_d$ contains fourth-order derivatives of $\phi_d$ (via $\nabla^2(\nabla^2 \phi_d)$). This makes the steady-state inverse problem effectively fourth order and a challenging testbed for stable high-order derivative learning.

**Setup.** We observe the final chromatin configuration (i.e., $\phi_h(x, y)$ and $\phi_e(x, y)$, equivalently $\phi_d(x, y)$) at $N_u$ spatial locations and infer the spatially varying reaction rate $\lambda(x, y)$ in the steady-state equation above. Training minimizes a data misfit on the observed chromatin fields together with the PDE residual enforced at $N_f$ collocation points. Native models compute the required spatial derivatives via autodiff, while mollified variants compute them via the Mollifier Layer (fixed convolutions), including the Laplacian and higher-order terms arising through $\nabla^2 \mu_d$. We report accuracy of $\lambda(x, y)$ (correlation) as well as intermediate derivative fidelity (e.g., Laplacian correlation where applicable) and computational cost (training time and peak memory; Table 2).

As shown in Fig 3b, mollified PINNs accurately estimate $\lambda(x, y)$ and stably recover the Laplacian as a key intermediate, unlike standard PINNs, which often misestimate its magnitude (see also Fig 1). Mollified models consistently outperform their native counterparts in both accuracy and efficiency (Table 2; additional qualitative comparisons across architectures are shown in Figs 21-24).

Fig 4a further shows that mollified PINNs capture spatial noise trends in $\lambda(x, y)$, which the native PINNs fail to achieve, highlighting improved recovery of heterogeneous parameter fields in this high-order setting. Additional implementation details and extended results are provided in App B.3.1 and Figs 21-24.

## 5 Biophysical Application

DNA architecture spans multiple spatial scales, with nanoscale features such as "heterochromatin domains" lying below the diffraction limit of conventional microscopy (Ricci et al. (2015)). Super-resolution techniques (Schermelleh et al. (2019)) like STORM (Rust et al. (2006)) overcome this barrier, enabling visualization of sub-diffraction chromatin structures and their spatial heterogeneity (Fig 4b). While such imaging provides rich qualitative insight, extracting mechanistic information demands integration with physical models.

Figure 4 summarizes this pipeline and its outputs. Fig 4a illustrates, on synthetic reaction-diffusion data, that mollified PINNs recover spatially heterogeneous noise trends in the inferred reaction rate field $\lambda(x, y)$,

Table 2: Comparison of baseline and mollified models across PDEs and metrics. **PINN**: Physics-Informed Neural Network, **PN**: PirateNet, **PF**: PINNsFormer, **FD**: FD-PINN, **+M** indicates the corresponding model augmented with mollifier layers.

| PDE Constraint (Order) | Metric | PINN | PINN+M | PN | PN+M | PF | FD |
|---|---|---|---|---|---|---|---|
| **1D Langevin ($1^{st}$)** | Time Taken (sec) | 2138 | **1615** | 2414 | **1473** | 6507 | 1527 |
| | Mean Corr. | **0.99** | 0.96 | 0.98 | **0.99** | 0.98 | 0.97 |
| | Temporal Corr. | 0.36 | **0.97** | **0.98** | 0.97 | 0.35 | 0.97 |
| | Peak Memory (GB) | 0.21 | **0.16** | 0.77 | **0.11** | 0.33 | 0.14 |
| **2D Heat ($2^{nd}$)** | Time Taken (sec) | 2294 | **1582** | 4737 | **2948** | 10200 | 1478 |
| | Mean Corr. | 0.81 | **0.99** | 0.13 | **0.22** | 0.73 | 0.82 |
| | Spatial Corr. | 0.21 | **0.99** | 0.16 | **0.20** | 0.04 | 0.76 |
| | Peak Memory (GB) | 1.20 | **0.24** | 1.15 | **0.11** | 0.80 | 0.21 |
| | Laplacian Corr. | 0.16 | **0.99** | 0.01 | **0.23** | 0.09 | 0.71 |
| **2D Reaction-Diffusion ($4^{th}$)** | Time Taken (sec) | 3386 | **335** | 817 | **125** | 34487 | 308 |
| | Mean Corr. | 0.44 | **0.99** | 0.77 | **0.99** | 0.68 | -0.73 |
| | Spatial Corr. | 0.17 | **0.84** | 0.25 | **0.91** | 0.47 | -0.86 |
| | Peak Memory (GB) | 2.75 | **0.23** | 0.48 | **0.12** | 1.90 | 0.21 |
| | Laplacian Corr. | 0.21 | **0.78** | 0.01 | **0.75** | 0.01 | -0.38 |

a key capability needed for robust inference under measurement variability. Fig 4b shows representative raw STORM images (Appendix C), which serve as the observational input to the inverse problem. Fig 4c reports the inferred reaction-rate statistics over image regions: the predicted mean and variance of $\lambda(x, y)$ closely track the corresponding ground-truth statistics in controlled synthetic benchmarks and remain spatially consistent with domain-like chromatin structure in STORM-derived fields.

Together, Fig 4a-c demonstrates that mollifier layers enable stable, high-order PDE-constrained inference on image-derived fields, supporting both accurate recovery of $\lambda(x, y)$ and faithful capture of spatial variability (mean/variance) that native PINNs miss in Sec 4.3. Resolving local epigenetic reaction rates from imaging reveals the mechanistic drivers of chromatin reorganization, linking nanoscale domain remodeling to gene regulation, cancer metastasis, and cell fate memory in development and disease contexts (Vinayak et al. (2025); Heo et al. (2023); Kant et al. (2024)). Our results thus establish mollifier layers as a robust tool for making PhiML scalable for integrating super-resolution imaging with biophysical modeling to extract interpretable parameters from high-dimensional biological data.

## 6 Discussion

By collapsing recursive higher-order automatic differentiation into a single analytic convolution, Mollifier Layers reduce both memory footprint and training time by 6–10×, as demonstrated in our experiments, while preserving robust, high-order derivative estimates. Though tested for inverse PDEs, the mollifier-based derivative layer extends naturally to forward solvers (Sec. B.5), operator learning, and neural ODEs where stable and accurate gradients are essential. This efficiency enables scalable physics-informed machine learning to tackle complex systems like kilometer-scale weather models (Bodnar et al. (2024); Palmer (2019)) or morphogenesis (Liu et al. (2024); Wyczalkowski et al. (2012)) simulations with markedly lower computational cost.

Limitations: Performance remains sensitive to the choice of mollifier kernel (see Sec. B.1.3), which must trade off noise suppression against high variance. The current hand-tuned implementation is limited near boundaries and on anisotropic grids. Developing adaptive or learned kernels, boundary-aware formulations, and validation strategies for adaptive meshes are key directions for future work.

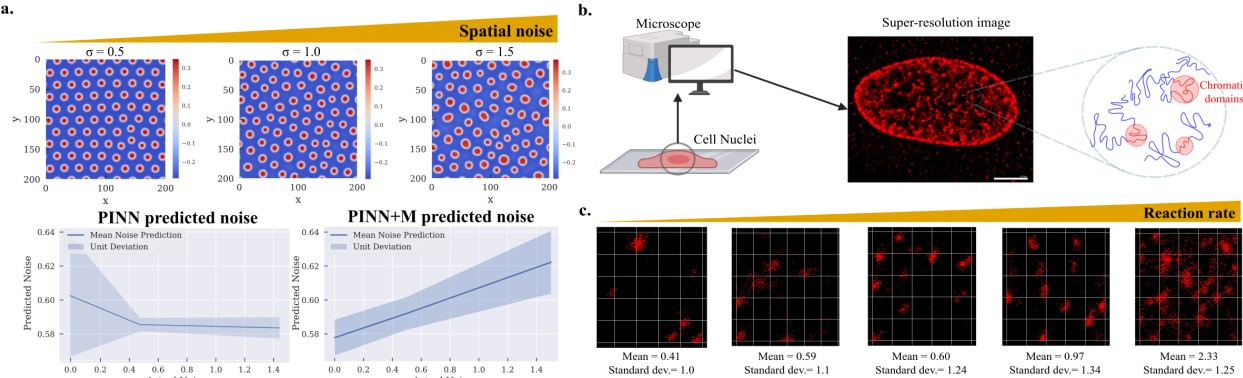

Figure 4: Mollified PINNs capture spatial heterogeneity in reaction-rate inference and enable PDE-constrained parameter extraction from STORM-derived chromatin fields. a) Synthetic benchmark: mollified PINNs recover spatial noise trends in the inferred reaction-rate field $\lambda(x, y)$. b) Example STORM image region preprocessed into a continuous chromatin-density field (Appendix C) used as input to the inverse problem. c) Predicted mean and variance of inferred reaction rates over image regions, showing faithful recovery of reaction-rate statistics and spatial consistency with chromatin domain structure.

## 7 Related Works

Recent physics-informed machine learning (PhiML) approaches—ranging from PINNs (Raissi et al. (2019)) and DeepONets (Lu et al. (2021)) to FNOs (Li et al. (2020)), PINOs (Li et al. (2024b)), and PIKANs (Patra et al. (2024))—have advanced PDE modeling. Our focus is on inverse problems and stable derivative computation, replacing recursive autodiff with a convolutional alternative. We review related work in these directions and outline our contributions.

Inverse Modeling: Recent work mitigates inverse-PhiML shortcomings in complementary ways. PINNverse (Almanstötter et al. (2025)) treats parameter recovery as a constrained optimization, improving robustness to noise; H-PINN (Chandrasukmana et al. (2025)) enforces Hadamard well-posedness to regularize ill-posed tasks; Bayesian PINNs (Yang et al. (2021)) place probabilistic priors on parameters to quantify uncertainty, albeit with significant sampling cost; and gPINNs (Yu et al. (2022)) augment the loss with gradient information to accelerate convergence; The PI-INN framework (Guan et al. (2024)) enables Bayesian inference via invertible networks but retain the full autodiff graph. Beyond the PINN family, operator-learning models such as the Latent Neural Operator (Wang & Wang (2024)) and Transformer-BVIP (Guo et al. (2022)) use cross-attention or boundary-aware transformers for inverse PDEs. Despite these advances, all rely on recursive autodiff. We replace it with a differentiable mollifier layer, enabling stable, memory-efficient, high-order derivatives while remaining plug-and-play.

Alternatives to Autodiff: Prior efforts to avoid recursive autodiff fall into three tracks: (i) finite-difference PhiML (Lim et al. (2022); Chiu et al. (2022); Huang & Alkhalifah (2024)) that save memory but lose high-order accuracy and suffers at high resolution, (ii) spectral PhiML (Maust et al. (2022); Patel et al. (2022)) that gain stability yet demand bespoke bases and (iii) operator-specific networks—e.g., Koopman operator nets (Lusch et al. (2018)), implicit Fourier/Green-function networks (Li et al. (2024a))—that embed analytical derivative kernels directly in the architecture; these deliver fast inference for a fixed PDE class but cannot generalize beyond the operators they hard-code. He et al. (2023) accelerate PINN training by estimating PDE derivatives through Monte Carlo sampling of Gaussian-smoothed networks using Stein's identity, replacing stacked back-propagation with stochastic expectation estimators. In contrast, our approach offers a general, architecture-agnostic alternative that enables stable, low-memory, high-order gradients—even in noisy, fourth-order PDEs—without sacrificing flexibility across PDE classes.

Alternative inverse-PDE paradigms: Beyond alternatives to autodiff for physics-informed training, there are complementary non-neural approaches to inverse PDEs. ODIL (Karnakov et al. (2024)) optimizes a discrete

physics/data loss over grid-valued unknown fields, whereas our mollifier layers retain a neural continuous surrogate and replace recursive autodiff with mollified convolutional derivative operators for stable high-order derivatives. Physics-Informed Regression (Nielsen et al. (2025)) estimates parameters via (often parameter-linear) regression/least-squares structure, whereas our approach is an architecture-agnostic derivative module that improves the efficiency and noise-robustness of high-order derivative evaluation in physics-informed neural models without assuming parameter-linearity.

## 8 Conclusion

Mollifier Layers offer a robust, scalable alternative to recursive autodiff, enabling stable high-order derivative estimation critical for inverse PDE learning. By decoupling differentiation from network depth, they unlock accurate, noise-resilient inference across complex physical systems. In doing so, Mollifier Layers pave the way for physics-informed machine learning to tackle previously intractable regimes—where deep models must reason stably over space, time, and noise—to extract scientific insight from data in real-world dynamical systems.

## Acknowledgments

This work was supported by NCI Award U54CA261694 (V.B.S.); NSF CEMB Grant CMMI-154857 (V.B.S.); NSF Grant DMS-2347834 (V.B.S.); NIBIB Awards R01EB017753 (V.B.S) and R01EB030876 (V.B.S.) and NIGMS Award R01GM155943 (V.B.S).

## Competing Interests

The authors declare no competing interests.

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

# A  Details of the models

## A.1  Details of the PINNs implementation

Significant progress has been made in improving PINNs since they first came out (Raissi et al. (2019)), specially showing that gradient flow can be significantly improved by adding residual connections (Wang et al. (2021a)). Additionally, it is well known that layer normalization helps with gradient smoothening and prevents explosion (Xu et al. (2019)). Since our aim is to accurately capture gradients and account for this progress made, we used an upgraded PINN, which has the following two features in addition to the base PINN architecture:

1. Residual Connections: Also known as *skip connections*, residual connections (He et al. (2016)) help mitigate the vanishing gradient problem and facilitate the training of deep neural networks. They can be incorporated into any layer by modifying the standard transformation in a multi-layer perceptron (MLP). Given an input $x$ to a node in a linear layer $L$, the standard output $f(x)$ is defined as:

$$f^L(x) = \phi(Wx + b) \tag{12}$$

   where $\phi$, $W$, and $b$ represent the activation function, weights, and biases, respectively. To introduce a residual (skip) connection between layer $L$ and layer $L + 1$, the modified output $f^L_{\text{res}}(x)$ is given by:

$$f^L_{\text{res}}(x) = \phi(Wx + b) + x \tag{13}$$

   This modification allows gradients to propagate more effectively through deeper layers, improving both convergence and model performance.

2. Layer Normalization: Layer normalization (Ba et al. (2016)) stabilizes and accelerates the training of deep neural networks by normalizing activations across features within each training example, rather than across the batch dimension as in batch normalization. This ensures consistent normalization even for small batch sizes, making it particularly useful in recurrent and transformer-based architectures. Given input features $x$, the output after applying layer normalization is:

$$f(x) = \frac{x - \mathbb{E}[x]}{\sqrt{\text{Var}(x) + \epsilon}} \gamma + \beta \tag{14}$$

   where $\mathbb{E}[x]$ and $\text{Var}(x)$ are the mean and variance computed over the feature dimension of each input, and $\gamma$ and $\beta$ are learnable scaling and shifting parameters. This transformation ensures stable feature distributions, improving training convergence and model performance.

## A.2  Details of the PirateNet implementation

We adopt the original architectures—with their core components intact—and fine-tune each one by varying their depth specifically for our problems, as detailed in Section B.

## A.3  Details of the PINNsFormer implementation

We adopt the original architectures—with their core components intact—and fine-tune each one by varying their depth specifically for our problems, as detailed in Section B.

## A.4  Common training choices across models

We use the following choices for all the models:

1. Fourier Features: As shown through previous works (Wang et al. (2021b)), input features mapped to a higher-dimensional space using *Fourier features* perform better in capturing high frequency

signals in the output. Given an input $x$, we define a set of frequencies $\omega_i$ within an interval $I \subset \mathbb{R}$. Specifically, we set $I = [-3, 3]$ and select $\omega_i$ at equal intervals within this range. The Fourier feature transformation then maps each input $x$ as follows:

$$x \rightarrow [\sin(\omega_1 x), \cos(\omega_1 x), \dots, \sin(\omega_n x), \cos(\omega_n x)] \tag{15}$$

where $\sin(x)$ and $\cos(x)$ are applied component-wise to each feature in $x$. This transformation expands the feature space, allowing the model to capture periodic patterns and high-frequency variations in the data. By incorporating both sine and cosine terms, Fourier features provide a richer oscillatory representation, improving the model's ability to approximate functions with diverse frequency components. This enhanced representation strengthens the model's ability to learn and generalize frequency-dependent behaviors.

2. Adam Optimization: Adam is an adaptive optimization algorithm (Kingma & Ba (2014)) widely used for training deep learning models. It combines the advantages of *Momentum* and *RMSprop*, adapting the learning rate for each parameter based on both the first moment (mean of gradients) and the second moment (uncentered variance of gradients). This adaptive mechanism allows Adam to efficiently navigate complex optimization landscapes, ensuring faster convergence, especially in problems involving sparse gradients or noisy datasets.

Adam maintains two moving averages for each parameter $\theta$:

- $m_t$: First moment (mean of gradients).
- $v_t$: Second moment (uncentered variance of gradients).

At time step $t$, these moments are updated as follows:

$$m_t = \beta_1 m_{t-1} + (1 - \beta_1) g_t \tag{16}$$

$$v_t = \beta_2 v_{t-1} + (1 - \beta_2) g_t^2 \tag{17}$$

where:

- $g_t$ is the gradient of the loss function with respect to $\theta$ at time step $t$.
- $\beta_1$ and $\beta_2$ are the exponential decay rates for the first and second moments, typically set to $\beta_1 = 0.9$ and $\beta_2 = 0.999$.

To correct the bias toward zero in the initial steps, Adam applies bias correction:

$$\hat{m}_t = \frac{m_t}{1 - \beta_1^t}, \quad \hat{v}_t = \frac{v_t}{1 - \beta_2^t} \tag{18}$$

Finally, the parameters are updated using:

$$\theta_t = \theta_{t-1} - \frac{\alpha}{\sqrt{\hat{v}_t} + \epsilon} \hat{m}_t \tag{19}$$

where:

- $\alpha$ is the learning rate.
- $\epsilon$ is a small constant (typically $10^{-8}$) to prevent division by zero.

Adam's adaptive learning rates make it effective across a wide range of problems, particularly those involving noisy or sparse gradients.

3. Cosine Learning Rate Decay (Loshchilov & Hutter (2016)): In deep learning, the learning rate is often initialized at a relatively high value and progressively reduced to refine the model and prevent overshooting minima. A common approach for this adjustment is *cosine annealing*, which schedules the learning rate to follow a cosine curve, ensuring a smooth and gradual decay. Compared to step decay, cosine decay has been shown to improve convergence and generalization.

The learning rate $\alpha_t$ at time step $t$ is defined as:

$$\alpha_t = \begin{cases} \eta, & \text{if } t < T_0 \\ \epsilon + \eta \cos\left(\frac{t - T_0}{T - T_0}\right), & \text{if } t \geq T_0 \end{cases} \tag{20}$$

where:

- $\eta$ is the initial learning rate.
- $\epsilon$ is the minimum learning rate (typically close to zero).
- $T$ is the total number of training iterations or epochs.
- $t$ is the current iteration or epoch.
- $T_0$ is the point at which the learning rate transitions from constant to decreasing.

This decay strategy ensures a smooth reduction in the learning rate, allowing the model to gradually settle into a well-generalized region of the loss landscape. When combined with *Adam optimization*, cosine annealing can lead to faster convergence and improved performance on unseen data.

## A.5   Choice of Mollifying function

With the constraints of *smoothness, compact support, and non-negativity* outlined in the main text, we define three mollifying functions to explore the behavior of our methodology. We consider a *second-order polynomial*, a *fourth-order polynomial*, and a *sine function* as potential mollification functions, along with the original *exponential mollifier kernel*. These functions are defined as follows:

Second-order polynomial:

$$f_2(x) = -(x^2 - R^2) \tag{21}$$

Fourth-order polynomial:

$$f_4(x) = -(x^2 - R^2)^2 \tag{22}$$

Sine function:

$$f_{\sin}(x) = \sin\left(\frac{\pi R x}{2} + \frac{\pi}{2}\right) \tag{23}$$

Original mollification function:

$$f_{\exp}(x) = e^{\left(\frac{-1}{1 - x^2}\right)} \tag{24}$$

where $x \in B_R(0)$, with $B_R(0)$ representing the $L_2$-norm ball centered at zero. We refer to $B_R(0)$ as $U$ in the main text for simplicity.

The kernel size is defined relative to the grid resolution, with values set as follows:

- Langevin equation: 0.01

- Heat equation: 0.005

- Reaction-diffusion system: 0.05

Thus, a kernel size of 7 corresponds to:

- $R = 0.3$ for the Langevin equation,

- $R = 0.015$ for the heat equation,

- $R = 0.15$ for the reaction-diffusion system.

We benchmarked the performance of these kernels using the *Langevin equation with a constant parameter with noise*, with details provided in **Section B.1**.

### A.6 Hardware specifications

Our computing core consists of nodes equipped with four Tesla K80 GPU cards, each containing two GK210GL GPUs, resulting in eight GPUs per node. The GK210GL GPU architecture features 13 Streaming Multiprocessors (SMs), with each SM comprising 192 CUDA cores, yielding a total of 2,496 CUDA cores per GPU. For each training run, we utilize a single node.

### A.7 Weak Form Motivation

The success of the Finite Element Method (FEM), which employs a weak form approach to solve Partial Differential Equations (PDEs) (Brenner (2008)Zienkiewicz et al. (2005)), has inspired the development of the mollification operation (Friedrichs (1944)) presented in this work. Lets begin with function $v, u, f : U \to R$. here u is the actual solution to PDE , f is the source term and $v$ is the weak form approximation of $u''$. In the weak form of a PDE, we aim to minimize the residual, which can be expressed as:

$$\int_U v(y)\phi(y)\,dy - \int_U f(y)\phi(y)\,dy = 0 \quad \forall\,\phi \in C_c^\infty(U) \tag{25}$$

We say $v = u''$ in the weak sense if,

$$\int_U v(y)\phi(y) = \int_U u''(y)\phi(y)\,dy = -\int_U u'(y)\phi'(y)\,dy \quad \forall\,\phi \in C_c^\infty(U) \tag{26}$$

This formulation helps us avoid the direct computation of higher-order derivatives, which would otherwise complicate the process. By approximating the PDE residual with derivatives, we can improve the accuracy of the solution. This leads us to propose an approach in which all derivatives are approximated in the weak form.

To achieve this, we employ a well-established approximation of functions in Sobolev spaces using mollification. This allows us to express the PDE residual in the weak form as:

$$\left| \int_U \eta_\epsilon(x - y)v(y)\,dy - \int_U \eta_\epsilon(x - y)f(y)\,dy \right| \tag{27}$$

Here, $v = D^\alpha u$ in the weak sense and $\eta_\epsilon(t)$ is a compact support bounded function in $(-\epsilon, \epsilon)$. Here $\eta$, also known as the mollifying function, is non-negative, infinitely differentiable and is defined as:

$$\eta_\epsilon(t) = \eta(\frac{t}{\epsilon}) \tag{28}$$

$$\eta(x) = e^{\frac{-1}{1-||x||_2^2}} \quad \forall ||x||_2 < 1 \tag{29}$$

$$\eta(x) = 0 \quad \forall ||x||_2 \geq 1 \tag{30}$$

We further relax the above residual by doing the following:

$$\left| \int_U \eta_\epsilon(x-y)\, v(y)\, dy - \int_U \eta_\epsilon(x-y)\, f(y)\, dy \right| \leq \left| \int_U \eta_\epsilon(x-y)\, v(y)\, dy - f(x) \right|$$
$$+ \left| f(x) - \int_U \eta_\epsilon(x-y)\, f(y)\, dy \right|. \tag{31}$$

Note that the residual part of our importance is:

$$\left| \int_U \eta_\epsilon(x-y) v(y) - f(x) \right| \tag{32}$$

The other term tends to zero as we decrease $\epsilon$:

$$\left| f(x) - \int_U \eta_\epsilon(x-y)\, f(y)\, \mathrm{d}y \right| = \left| \int_U \eta_\epsilon(x-y)\big(f(x) - f(y)\big)\, \mathrm{d}y \right|$$
$$\leq \frac{1}{\epsilon^n} \int_{B(x,\epsilon)} \eta\!\left(\tfrac{x-y}{\epsilon}\right) \big|f(x) - f(y)\big|\, \mathrm{d}y. \tag{33}$$

$$\frac{1}{\epsilon^n} \int_{B(x,\epsilon)} \eta\left(\frac{x-y}{\epsilon}\right) |f(x) - f(y)|\, dy \leq \frac{C}{\epsilon^n} \int_{B(x,\epsilon)} |f(x) - f(y)| \tag{34}$$

where $C = \sup_{t \in \mathbb{R}^n} \eta(t)$. Using the Lebesgue Differentiation Theorem, we can state that as $\epsilon \to 0$:

$$\frac{1}{\epsilon^n} \int_{B(x,\epsilon)} |f(x) - f(y)|\, dy \to 0 \tag{35}$$

Now, we can see that $v$ is our weak derivative (approximated by a neural network). Thus, the expression becomes:

$$u_{\mathrm{pred}} = \eta_\epsilon * \mathrm{NN}_\theta \tag{36}$$

Here $\mathrm{NN}_\theta$ is the final output form our Neural network layers.

As we are working on the grid (G), so we can express the PDE residual with the help of Mollifying operations as:

$$\frac{\left( \sum_{x \in G} D_x^\alpha \eta_\epsilon * \mathrm{NN}_\theta(x) - f(x) \right)^2}{|G|} \tag{37}$$

Here $x$ is sampled randomly from a predefined grid of points in the domain.

## A.8 Convergence Analysis of the Mollifier Layer

Let $f : [0,1] \to \mathbb{R}$ be continuously differentiable and $L$-Lipschitz. Denote the uniform grid spacing by $h > 0$ and the kernel half-width by $\delta > 0$. For a nonnegative bump $\rho \in C_c^\infty([-1,1])$ with unit mass, define the mollifier

$$(J_\delta f)(x) = \int_{-\delta}^{\delta} \rho_\delta(r) f(x+r)\, dr, \qquad \rho_\delta(r) = \delta^{-1}\rho(r/\delta). \tag{38}$$

Measured data are $g_j = f(x_j) + \eta_j$ at $x_j = jh$ with $|\eta_j| \leq \varepsilon$. Throughout, $C, C_1, C_2, \ldots$ denote positive constants independent of $\delta$, $h$, and $\varepsilon$.

Mean-square consistency. (Murio et al. (1998)) For $f \in L^2([0,1])$,

$$\|J_\delta f - f\|_{L^2} \to 0 \quad \text{as } \delta \to 0. \tag{39}$$

Uniform consistency (Murio et al. (1998)).

$$\|J_\delta f - f\|_\infty \le CL\delta, \qquad \|\partial_x J_\delta f - f'\|_\infty \le CL\delta. \tag{40}$$

Discrete mollification (Murio et al. (1998)).

$$\|J_\delta g - f\|_\infty \le C(\delta + h + \varepsilon). \tag{41}$$

Finite-difference accuracy (Murio et al. (1998)).

$$\left| D_0(J_\delta g) * j - \partial_x J_\delta f(x_j) \right| \le Ch^2, \quad D_0 g_j = \frac{g_{j+1} - g_{j-1}}{2h}. \tag{42}$$

Uniform derivative consistency (Murio et al. (1998)).

$$\|D_0(J_\delta g) - f'\|_\infty \le C(\delta + h^2 + \varepsilon). \tag{43}$$

Choosing $\delta \approx \sqrt{h}$ and $h^2 \approx \varepsilon$ minimises the bound, giving $\mathcal{O}(\sqrt{h})$ uniform error while the backward operator norm $\|\partial_x J_\delta\|$ remains $\mathcal{O}(1/\delta)$.

For higher-order derivatives, differentiating under the integral sign yields

$$\partial_x^n(J_\delta f) = J_\delta^{(n)} f, \tag{44}$$

where the superscript acts on the kernel. The same arguments with a centred stencil of length $2n$ give

$$\|\partial_x^n(J_\delta f) - f^{(n)}\|_\infty \le C_n(\delta + h + \varepsilon). \tag{45}$$

Finally, in the case $n = 1$ and absorbing $h^2$ into the constants we obtain

$$|\partial_x(J_\delta f) - f'|_\infty \le C_1 \delta + C_2(h + \varepsilon), \tag{46}$$

which is the bound quoted in the main text.

**Comparison with finite-difference (FD) derivatives.** To clarify the distinction between finite-difference (FD) schemes and mollifier layers, we summarize both the theoretical error behavior and the empirical outcomes observed in our experiments.

*Finite differences.* FD schemes approximate derivatives by forming weighted sums of nearby grid values. For higher-order derivatives, the associated coefficients become increasingly oscillatory and extreme (see Fornberg (1988); Lele (1992)), which amplifies any measurement noise. Analytically, for derivative order $m$ on a grid with spacing $h$, the FD error bound includes a noise-amplification term of order $\varepsilon/h^m$, in addition to truncation error terms $O(h^p)$ depending on the scheme order $p$. Thus, as $m$ grows, noise is magnified by the factor $1/h^m$, rendering high-order FD estimates unstable.

*Mollifier layers.* In contrast, mollifier-based differentiation first convolves the signal with a smooth, compactly supported kernel before differentiation. This convolution damps high-frequency components, preventing noise amplification. As derived, the derivative error bound for mollifiers takes the form

$$\|D_0(J_\delta g) - u'\|_\infty \le C_1 \delta + C_2(h + \epsilon),$$

Importantly, the growth with derivative order is absorbed into the mollifier kernel, so stability is preserved even for high-order PDEs.

*Empirical validation.* Our experiments confirm this theoretical distinction:

- For the 1D Langevin equation (first order), FD-PINNs and mollifier-PINNs perform comparably.

- For the 2D heat equation (second order), FD-PINN accuracy deteriorates, while mollifier-PINNs remain stable.

- For the 2D reaction–diffusion equation (fourth order), FD-PINNs collapse entirely, whereas mollifier-PINNs retain high accuracy.

Together, both the analytic bounds and experimental results demonstrate that FD-based differentiation suffers from noise amplification that scales unfavorably with derivative order, while mollifier layers—by acting as a built-in low-pass filter—maintain stable, accurate performance in high-order and noisy settings.

## B  Application-wise details

### B.1  Langevin equation

#### B.1.1  Training Setup for All Models

PINN architechures: The PINN architecture for predicting $\hat{u}$ consists of seven layers, each with 1000 nodes and ReLU activation functions. To enhance temporal encoding, we augment the input time $t$ with Fourier features (details in appendix A). The model is trained using the Adam optimizer with a cosine annealing learning rate schedule. The corresponding mollified PINN maintains the same architecture and training epochs but includes the mollifying layer.

PirateNet arch: PirateNet models the stochastic ordinary differential equation $u_t = u + \lambda(t)$ with an eight-layer fully connected residual network, each layer having 256 hidden units and tanh activations. The scalar time input $t$ is first lifted into a 256-dimensional Fourier feature embedding (scale factor 2) to capture high-frequency temporal structure. Each layer applies a custom *PI Modified Bottleneck* block: a three-layer MLP whose residual update is mixed via two learnable nonlinear functions $u(\cdot)$ and $v(\cdot)$, and whose skip connection is scaled by a trainable parameter $\alpha$. Finally, two parallel output heads predict the solution $\hat{u}(t)$ and the auxiliary forcing coefficient $\hat{\lambda}(t)$, enabling the network to learn both the state evolution and the driving noise in one joint framework.

PINNsFormer arch: When applied to the Langevin equation $u_t = u + \lambda(t)$: PINNsformer is configured as a four-layer transformer encoder–decoder with model dimension 64 and four attention heads per layer. The input time $t$ is first encoded using learned sinusoidal embeddings. Each transformer layer applies multi-head self-attention followed by a three-layer feed-forward block activated by WaveAct, a learnable mixture of $\sin(x)$ and $\cos(x)$. Full residual connections and layer normalization are maintained throughout. Two decoder heads then output the state $\hat{u}(t)$ and the auxiliary estimate $\hat{\lambda}(t)$, allowing the attention-based model to jointly learn deterministic evolution and stochastic forcing.

All models are trained on a grid sampled from the interval $[0, 1]$, using a batch size of 200 for 1000 epochs. The initial learning rate is set to $10^{-3}$ with cosine decay scheduling. The Adam optimizer is configured with parameters $\beta_1 = 0.9$, $\beta_2 = 0.999$.

#### B.1.2  Evaluation on the Langevin Equation

To test the models, we first infer a constant forcing term for the Langevin equation. As shown in Fig. 5-7, all models successfully predict the constant forcing term. We then extend our analysis to Gaussian white noise predictions, with results summarized in Section 4.1 and Table 2. The figures for all the models are Fig. 8-10. Mollified versions outperform the native nets in all cases.

Thereafter, we also test the models on temporally varying signal along with its noisy version Fig. 13-16. Here too the mollified models outperform the native models.

#### B.1.3  Impact of Functional Choices

The effectiveness of Mollifier-Nets depends on two key parameters:

1. The order of the mollifying function $\eta$.

2. The kernel size, i.e., the integration domain $U$.

For the Langevin equation with constant noisy parameter, various configurations accurately estimate the mean $\Lambda$ (Fig. 11), but they differ in their ability to capture noise variations.

Kernel Size Effect: As shown in Fig. 11, for a second-order polynomial mollifier, increasing the kernel size degrades performance on large noise, while a small kernel size fails to capture finer noise variations. An optimal kernel size of 10-15 balances noise capture effectively. This behavior can be attributed to the frequency distribution of the underlying function.

Mollifier Order Effect: As shown in Fig. 12, higher-order polynomials perform worse at capturing noise compared to lower-order mollifying functions.

Conclusively, both polynomial order and kernel size can be optimized. While a low-order polynomial with a moderate kernel size is ideal for extracting Gaussian white noise in the Langevin setting, these choices may vary for different learnable functions.

## B.2 Heat equation

TRAINING SETUP

PINN architecture: The neural network consists of 10 layers, each with 250 nodes and tanh activation functions. Spatial Fourier features are used for positional encoding. Training is performed using the Adam optimizer with a learning rate that progressively decreases following a cosine decay schedule. The mollified version consists of the mollifier layer in addition to the base arch with a original mollifying function.

PirateNet architecture:To solve the two-dimensional heat equation, PirateNet uses the same eight residual layers with 256 units each and tanh nonlinearities. The spatial coordinates $(x, y)$ are first transformed via a 256-dimensional Fourier embedding (scale 2) to enrich spatial frequencies. Inside each PI Modified Bottleneck block, the network mixes the three-layer MLP update with $u(\cdot)$ and $v(\cdot)$ transforms, and applies the learned $\alpha$ scaling to the skip path. The main output head produces the temperature field $\hat{u}(x, y)$, while an auxiliary head estimates the spatial diffusivity map $\hat{\lambda}(x, y)$, jointly recovering both solution and physical parameter.

PINNsFormer architecture: To handle two-dimensional heat flow, PINNsformer is simplified to a single transformer layer of dimension 32 with one attention head. The spatial coordinates $(x, y)$ are mapped via learned 2D sinusoidal embeddings before entering a WaveAct-activated feed-forward block. Residual connections and normalization persist in this minimal transformer. Its two output heads predict the temperature field $\hat{u}(x, y)$ and the diffusivity field $\hat{\lambda}(x, y)$, embedding both solution and parameter estimation within a physics-aware attention framework.

The models are trained on a grid-sampled interval with a batch size of 2500 for 1000 epochs. The initial learning rate is set to $10^{-3}$, with Adam optimizer parameters $\beta_1 = 0.9$, $\beta_2 = 0.999$.

The results are quantitatively summarized in table 2 and the plots are Fig. 18-20.

## B.3 Reaction Diffusion equation

### B.3.1 Reaction-Diffusion Model for Chromatin Organization

The spatial organization of chromatin within the nucleus plays a fundamental role in gene regulation, influencing cellular function and fate (Gilbert et al. (2005)). Chromatin exists in two primary states: heterochromatin, which is compacted and transcriptionally inactive, and euchromatin, which is more open and transcriptionally active. The dynamic interplay between these states is governed by diffusion and biochemical reactions, particularly modifications such as methylation and acetylation (Kant et al. (2024); Vinayak et al. (2025)).

Mathematical Model:

To model this system, the authors (Kant et al. (2024)) consider the nucleus as a mixture of three components: nucleoplasm ($\phi_n$), euchromatin ($\phi_e$), and heterochromatin ($\phi_h$), where their volume fractions satisfy the space-filling constraint:

$$\phi_h(x,t) + \phi_n(x,t) + \phi_e(x,t) = 1. \tag{47}$$

To simplify the system, they define two independent variables:

- Nucleoplasm volume fraction ($\phi_n(x,t)$): Represents the proportion of nucleoplasm at a given location.

- Chromatin phase difference ($\phi_d(x,t)$): Defined as the difference between heterochromatin and euchromatin fractions:

$$\phi_d(x,t) = \phi_h(x,t) - \phi_e(x,t). \tag{48}$$

The order parameter $\phi_d(x,t)$ indicates whether a region is euchromatin-rich ($\phi_d < 0$) or heterochromatin-rich ($\phi_d > 0$).

Free Energy Formulation: The free energy density $F(x,t)$ of the system is given by:

$$F(x,t) = \left[\phi_e^2 + \phi_h^2(\phi_h^{\mathrm{max}} - \phi_h)^2\right] + \frac{\delta^2}{2}|\nabla\phi_n|^2 + \frac{\delta^2}{2}|\nabla\phi_d|^2, \tag{49}$$

where:

- The first term represents chromatin-chromatin interactions, modeled using a Flory-Huggins-type potential with minima at $\phi_h = 0$ (euchromatin) and $\phi_h = \phi_h^{\mathrm{max}}$ (heterochromatin).

- The second term accounts for interfacial energy, penalizing sharp transitions between chromatin phases and promoting smooth boundaries. The parameter $\delta$ controls the interface width.

Diffusion Kinetics of Nucleoplasm: The diffusion of nucleoplasm ($\phi_n$) is governed by the gradient of the chemical potential $\mu_n(x,t)$, which drives the system toward a lower-energy configuration. The evolution of nucleoplasm follows the diffusion equation:

$$\frac{\partial\phi_n}{\partial t} = \nabla^2\mu_n, \tag{50}$$

where $\nabla^2$ is the Laplacian operator, ensuring a conservative dynamics where total nucleoplasm content remains constant unless boundary conditions specify otherwise.

Reaction-Diffusion Model for Histone Modifications: Epigenetic modifications such as methylation and acetylation mediate the conversion between euchromatin and heterochromatin. The evolution of the chromatin phase difference ($\phi_d$) is governed by:

$$\frac{\partial\phi_d}{\partial t} = \nabla^2\mu_d + 2(\lambda\phi_e - \phi_h), \tag{51}$$

where:

- $\nabla^2\mu_d$ represents the diffusive spread of epigenetic modifications.

- $\lambda\phi_e - \phi_h$ models methylation-driven conversion from euchromatin to heterochromatin.

The chemical potential $\mu_d$ associated with this transition is:

$$\mu_d = -\phi_e + \phi_h(\phi_h^{\mathrm{max}} - \phi_h)(\phi_h^{\mathrm{max}} - 2\phi_h) - \delta^2\nabla^2\phi_d. \tag{52}$$

Thus, combining equation 51 and 52 yields a system which is governed by a fourth-order differential equation, indicating complex spatiotemporal dynamics.

Numerical Solution and Simulations: To numerically solve the system, we discretize the evolution equations for $\phi_n$ and $\phi_d$, along with their corresponding chemical potentials $\mu_n$ and $\mu_d$. The simulations assume no net exchange of chromatin or nucleoplasm with the surroundings, though boundary conditions can be adjusted to model specific biological scenarios.

### B.3.2   Training setup:

PINN architechure: The neural network comprises 10 layers, each with 250 nodes using a tanh activation function. We use spatial Fourier features as positional encoding. Training is performed using the Adam optimizer, with a learning rate that progressively decreases during training.

PirateNet architechure:

**PirateNet for the reaction–diffusion system.** For the reaction–diffusion PDE, PirateNet remains an eight-layer tanh MLP with 256-unit width. The inputs $(x, y)$ pass through the same 256-dimensional Fourier embedding before entering successive PI Modified Bottleneck blocks, each mixing residual updates via learnable $u$ and $v$ functions and scaling skips by $\alpha$. Two output heads then predict the concentration field $\hat{\phi}_d(x, y)$ and the reaction-rate field $\hat{\lambda}(x, y)$, capturing both diffusion dynamics and reaction kinetics in one cohesive residual architecture.

PINNsFormer architechure: For reaction-diffusion, PINNsformer again uses one transformer layer of size 32 with a single attention head. Inputs $(x, y)$ receive learned sinusoidal embeddings and pass through a WaveAct feed-forward subblock, all under full residual connections and layer normalization. The model's main head outputs the concentration field $\hat{\phi}_d(x, y)$, while an auxiliary head outputs the reaction-rate map $\hat{\lambda}(x, y)$, integrating physics constraints directly into the attention-based architecture.

The models were trained on a grid sampled on the interval using a batch size of 2500 for 500 epochs. A learning rate of $1 \times 10^{-3}$ with cosine decay scheduling was used. Training employed the Adam optimizer with specific parameters: $\beta_1 = 0.9$, $\beta_2 = 0.999$.

### B.4   Correlation calculations, computing time estimates and memory efficiency estimates

We assess the accuracy and computational efficiency of our model by measuring the correlation between the actual and predicted parameter values, laplacians and the time/memory usage respectively. These evaluations ensure that our method is not only effective but also feasible for real-world applications where resources such as time and memory are constrained.

### B.4.1   Mean and Noise Calculation

To evaluate the robustness of our model and assess its sensitivity to statistical fluctuations, we conduct multiple runs on the same problem. Each run produces two key metrics: the predicted mean and the predicted noise.

Predicted Mean: The model outputs an estimate of the central tendency of the target variable for each run. This predicted mean represents the model's best approximation of the expected output given the input features. By averaging the predicted means across multiple runs, we obtain a consolidated estimate of the model's central tendency.

Predicted variance: In addition to the predicted mean, we compute the variability or noise associated with each prediction. By averaging these predicted noise values, we gain insights into the uncertainty of the model's predictions.

Correlation Analysis: Once the actual and predicted mean, variance and the laplacians are computed, the next step is to assess the correlation between the actual and predicted values. This ensures that the model effectively captures underlying trends and relationships in the data.

To quantify this relationship, we compute the **correlation coefficient**, defined as:

$$r = \frac{\sum\limits_{i=1}^{n}(X_i - \bar{X})(Y_i - \bar{Y})}{\sqrt{\sum\limits_{i=1}^{n}(X_i - \bar{X})^2 \sum\limits_{i=1}^{n}(Y_i - \bar{Y})^2}} \tag{53}$$

where:

- $X_i$ and $Y_i$ represent the actual and predicted values, respectively.

- $\bar{X}$ and $\bar{Y}$ denote the mean of the actual and predicted values.

This correlation coefficient provides a quantitative measure of the strength and direction of the relationship between actual and predicted values.

### B.4.2 Time Evaluation

To measure the execution time of our model, we use Python's built-in `time` module, which provides a simple way to capture the start and end times of a computation. By recording the time before and after running the model, we can compute the total time taken for a single run. This approach allows us to evaluate the speed of our model under various conditions and for different input sizes.

Additionally, execution time measurements across different experiments enable the identification of performance bottlenecks and an assessment of whether the model's time complexity scales appropriately with increasing data or model size.

### B.4.3 Memory Evaluation

To measure memory consumption, we utilize PyTorch's built-in utilities for tracking memory usage during training and inference. Specifically, when using GPUs, we employ the functions `torch.cuda.memory_allocated()` and `torch.cuda.memory_reserved()` to monitor memory allocation.

By tracking both GPU and system memory consumption, we can determine whether the model is memory-efficient and ensure that it operates within available system resources without causing memory overflow or excessive usage.

### B.5 Forward toy problem

While our focus is on inverse problems, as demonstrated extensively throughout this work, we briefly illustrate the applicability of mollifier layers to a forward problem. The forward problem is governed by the linear ordinary differential equation $\frac{d}{dx}u(x) + \lambda u(x) = 0$, with $\lambda = 1.0$. The solution $u(x)$ is approximated using a fully connected neural network with hidden layer of 64 neurons each and Tanh activations. To compute spatial derivatives in a stable and differentiable manner, we employ mollifier-based convolutions. The loss function consists of two terms: an ODE loss that enforces the differential equation, and a data loss that aligns the mollified prediction with the sampled data. We use a mollifier kernel of size 11 with a smoothing radius $\varepsilon = 5\Delta x$, where the spatial discretization $\Delta x = 0.05$. The model is trained using the Adam optimizer with cosine learning rate decay over 5000 epochs. This example illustrates how mollifier layers can be used in the forward problem. The predictions as well as the learning curves are provided in Fig. 26.

## C  Super-resolution image processing

Since the super-resolution images are binary, they only indicate the presence or absence of a signal at a given locus. The density of these activated loci reflects the chromatin density at a given position, where a higher density corresponds to more heterochromatin. To establish a correspondence between the imaging

data and the reaction-diffusion system, it is necessary to determine a density equivalent to the obtained super-resolution imaging data. In this work, we define a new square grid to better identify the distribution of chromatin within the nucleus. To compute the density of chromatin at a given point on this grid, we use the following methodology:

Watson Kernel interpolation (Watson (1964)) is employed to perform spatial interpolation of data points in super-resolution images. This method utilizes a Gaussian kernel to compute a weighted average estimate of nearby points, ensuring smooth interpolation over the image grid. The Gaussian kernel is defined as:

$$K(d) = \frac{\exp\left(-\frac{d^2}{2\sigma^2}\right)}{\sqrt{2\pi\sigma^2}} \tag{54}$$

where $d$ represents the Euclidean distance between points, and $\sigma$ is the kernel's bandwidth parameter, which controls the kernel width. The kernel assigns higher weights to closer points, effectively smoothing the interpolation based on proximal density.

Then the Nadaraya-Watson estimator (Nadaraya (1964); Watson (1964)) is used to compute the weighted average of data values, where the weights are determined by the above Gaussian kernel. Specifically, for a query point $(x_{\text{new}}, y_{\text{new}})$, the estimated value $z_{\text{new}}$ is calculated as:

$$z_{\text{new}} = \frac{\sum_{i=1}^{n} w_i z_i}{\sum_{i=1}^{n} w_i} \tag{55}$$

where $w_i$ is the weight assigned to each data point based on its distance to $(x_{\text{new}}, y_{\text{new}})$, and $z_i$ is the corresponding value at each data point.

For computational efficiency, *batch processing* is used to perform interpolation in chunks. This method divides the target grid into smaller batches and computes the weighted average for each batch, ensuring that the interpolation process scales efficiently with large datasets. The interpolation is implemented using the following steps:

1. Compute the distances between target points and known data points.

2. Apply the Gaussian kernel to compute weights.

3. Calculate the weighted sum of the data values.

4. Normalize the weighted sum by the total weights to obtain the interpolated values.

After applying Watson Kernel interpolation, the interpolated values are used to update image features such as $\phi_h$, $\phi_e$, and $\phi_n$. These terms are computed as:

$$\phi_h = \left(\frac{z_{\text{mesh}} \times 6}{7}\right)^{0.5}, \tag{56}$$

$$\phi_n = \left(\frac{z_{\text{mesh}} - 0.7}{0.7}\right)^2 \times \frac{6}{7 \times 0.7} + 0.2, \tag{57}$$

$$\phi_e = 1 - \phi_h - \phi_n. \tag{58}$$

These processed values are then used for further analysis, providing refined estimates of the underlying super-resolution imaging features.

# D  Extension to Multiple Unknown Parameters / Fields

This appendix formalizes how our mollifier-layer framework extends from a single unknown parameter field to the case of multiple unknown parameters (or multiple coupled fields). The key point is that mollifier layers define a *derivative operator* applied at the output, and thus extend component-wise to any number of predicted quantities.

## D.1  Multi-parameter inverse-PDE formulation

Consider a PDE constraint written abstractly as

$$\mathcal{F}\Big(t, x, \ u(t,x), \ \{\partial^\alpha u(t,x)\}_{|\alpha|\leq k}, \ \boldsymbol{\lambda}(t,x)\Big) = 0, \tag{59}$$

where $u : \mathcal{T} \times \Omega \to \mathbb{R}^{d_u}$ is the (possibly vector-valued) state, and $\boldsymbol{\lambda}(t,x) = (\lambda_1(t,x), \ldots, \lambda_P(t,x)) \in \mathbb{R}^P$ collects $P$ unknown parameters/fields to be inferred. The PDE residual typically requires derivatives of $u$ up to order $k$.

## D.2  Network outputs and mollifier layer (component-wise)

In our method, the base model predicts a pre-mollified output $\hat{g}$ and the mollifier layer produces the smoothed field $\hat{u}$ by convolution with a compactly supported mollifier $\eta$:

$$\hat{u}(n) = (\hat{g} * \eta)(n) = \int_{m\in U} \hat{g}(m)\, \eta(n-m)\, dm, \tag{60}$$

and derivatives are computed by convolving $\hat{g}$ with analytic derivatives of $\eta$:

$$\partial_j \hat{u}(n) = (\hat{g} * \partial_j \eta)(n), \qquad \partial^\alpha \hat{u}(n) = (\hat{g} * \partial^\alpha \eta)(n), \tag{61}$$

as in Eqs. (4–6) of the main paper.

For multiple unknown parameters, we predict them as additional outputs of the model. A convenient parameterization is

$$(t,x) \mapsto \big(\hat{g}(t,x), \ \hat{\boldsymbol{\lambda}}(t,x)\big), \tag{62}$$

where $\hat{\boldsymbol{\lambda}}$ may be (i) predicted directly, or (ii) predicted via its own pre-mollified outputs and (optionally) mollified in the same way as $\hat{u}$ if one desires the same smoothing/regularization:

$$\hat{\lambda}_p(n) = (\hat{g}_{\lambda_p} * \eta)(n), \quad p = 1, \ldots, P \quad \text{(optional)}. \tag{63}$$

Importantly, these equations are applied component-wise when $u$ is vector-valued; no change is required beyond carrying extra output channels.

## D.3  Residual and training objective

Using mollifier-based derivatives, we define the residual at collocation points $(t_i, x_i)$:

$$\hat{f}(t_i, x_i) = \mathcal{F}\Big(t_i, x_i, \ \hat{u}(t_i, x_i), \ \{\partial^\alpha \hat{u}(t_i, x_i)\}_{|\alpha|\leq k}, \ \hat{\boldsymbol{\lambda}}(t_i, x_i)\Big). \tag{64}$$

Training then minimizes the same PhiML objective as in the single-parameter case, with $\hat{\boldsymbol{\lambda}}$ replacing $\hat{\lambda}$:

$$\text{MSE}_{\text{total}} = \text{MSE}_u + \text{MSE}_f, \qquad \text{MSE}_u = \frac{1}{N_u}\sum_{i=1}^{N_u} \|\hat{u}(t_i, x_i) - u(t_i, x_i)\|^2, \qquad \text{MSE}_f = \frac{1}{N_f}\sum_{j=1}^{N_f} \|\hat{f}(t_j, x_j)\|^2. \tag{65}$$

If supervision or priors are available for any parameter fields, one can add additional terms (e.g., $\|\hat{\lambda}_p - \lambda_p\|^2$ or smoothness regularizers), but these are not required for the extension.

### D.4 Remarks

- **What changes with $P > 1$.** Only the dimensionality of the parameter output increases from $\hat{\lambda}$ to $\hat{\boldsymbol{\lambda}}$; mollifier derivative computation for $\hat{u}$ is unchanged.

- **When to mollify parameters.** If the PDE uses derivatives of parameters (or one wants additional smoothing/regularization on $\hat{\boldsymbol{\lambda}}$), apply the same mollifier construction to $\hat{\lambda}_p$ via Eq. equation 63; otherwise predicting $\hat{\boldsymbol{\lambda}}$ directly is sufficient.

- **Separable/closed-form parameter recovery.** For PDEs where parameters can be expressed algebraically from predicted quantities (as discussed in Sec. 3.3), the same idea applies component-wise to each parameter in $\boldsymbol{\lambda}$ whenever such a separable form exists.

## E  Code and data availability

Code and data generated: All the code generated, along with the required data for the paper has been stored here: `https://doi.org/10.5281/zenodo.15420249`

STORM imaging data availability: We have provided the data for the 5 sections for which the parameters have been calculated with the uploaded data. A full sample image can be found at the Dhankhar et al. (2025) author's github repository at: `https://github.com/ShenoyLab/STORM_Analysis`

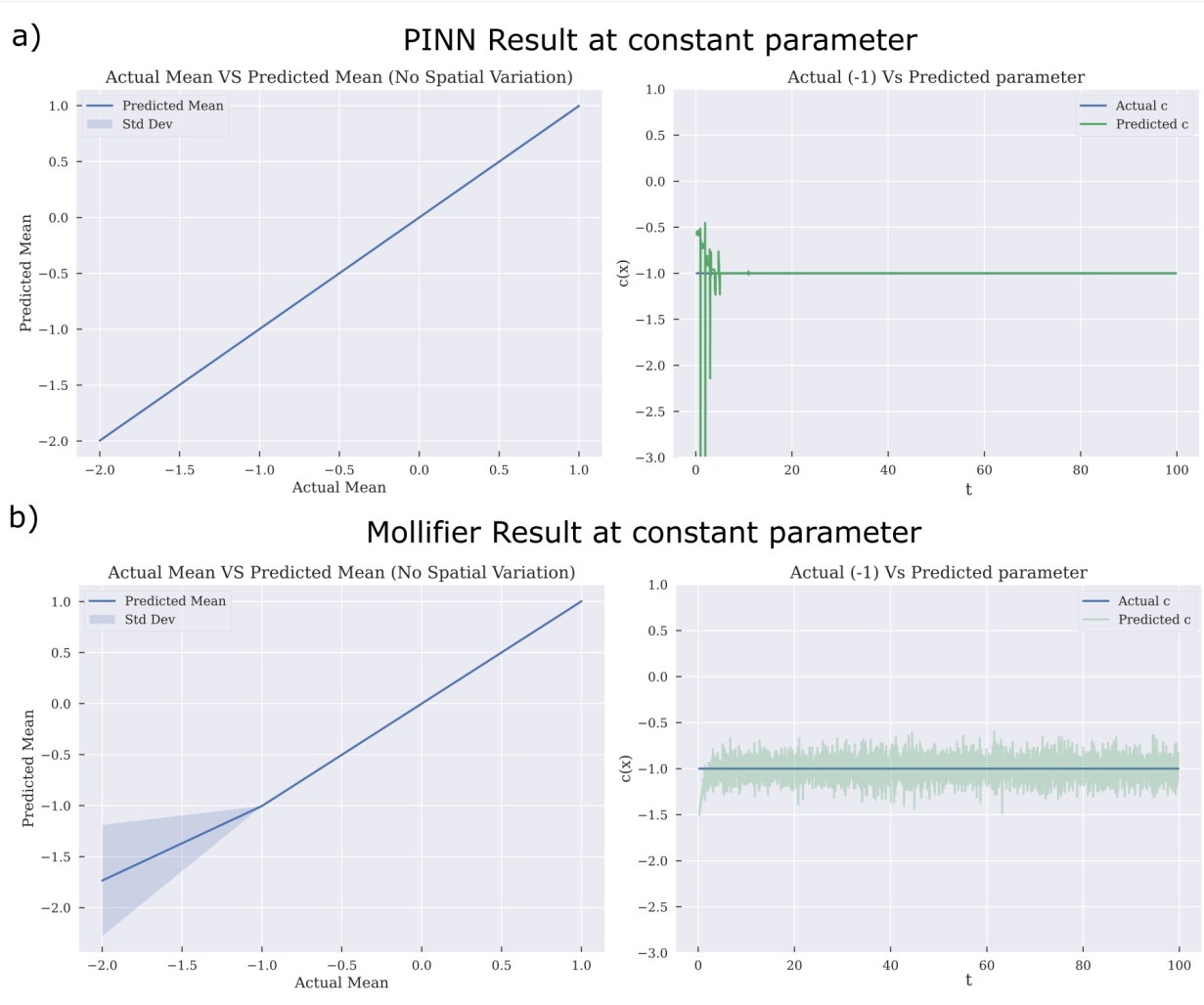

Figure 5: *Constant forcing term inference in Langevin equation using PINN and Mollified PINN*

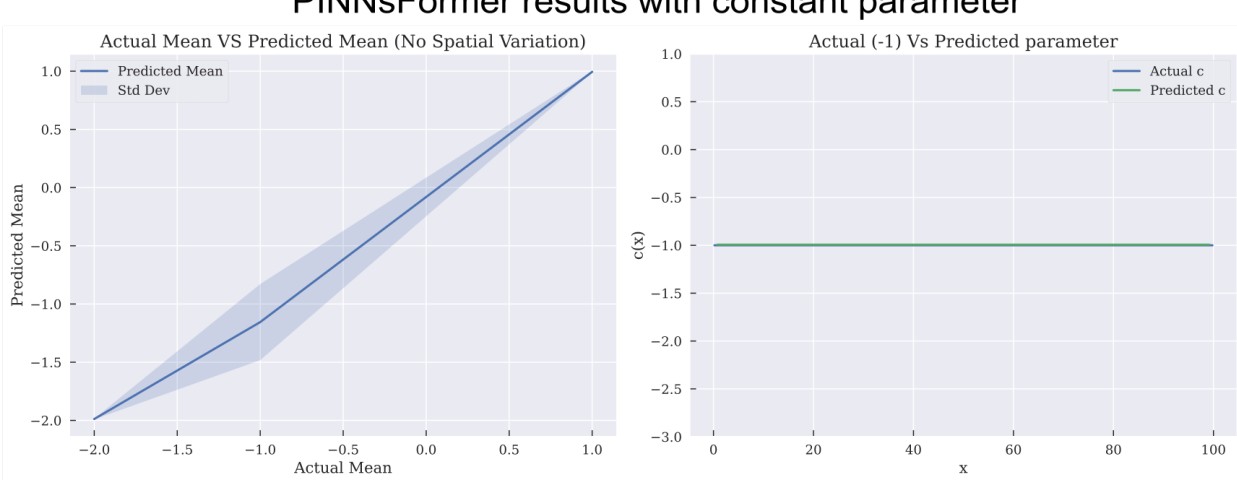

Figure 6: *Constant forcing term inference in Langevin equation using PirateNet and Mollified PirateNet*

Figure 7: *Constant forcing term inference using PINNsFormer*

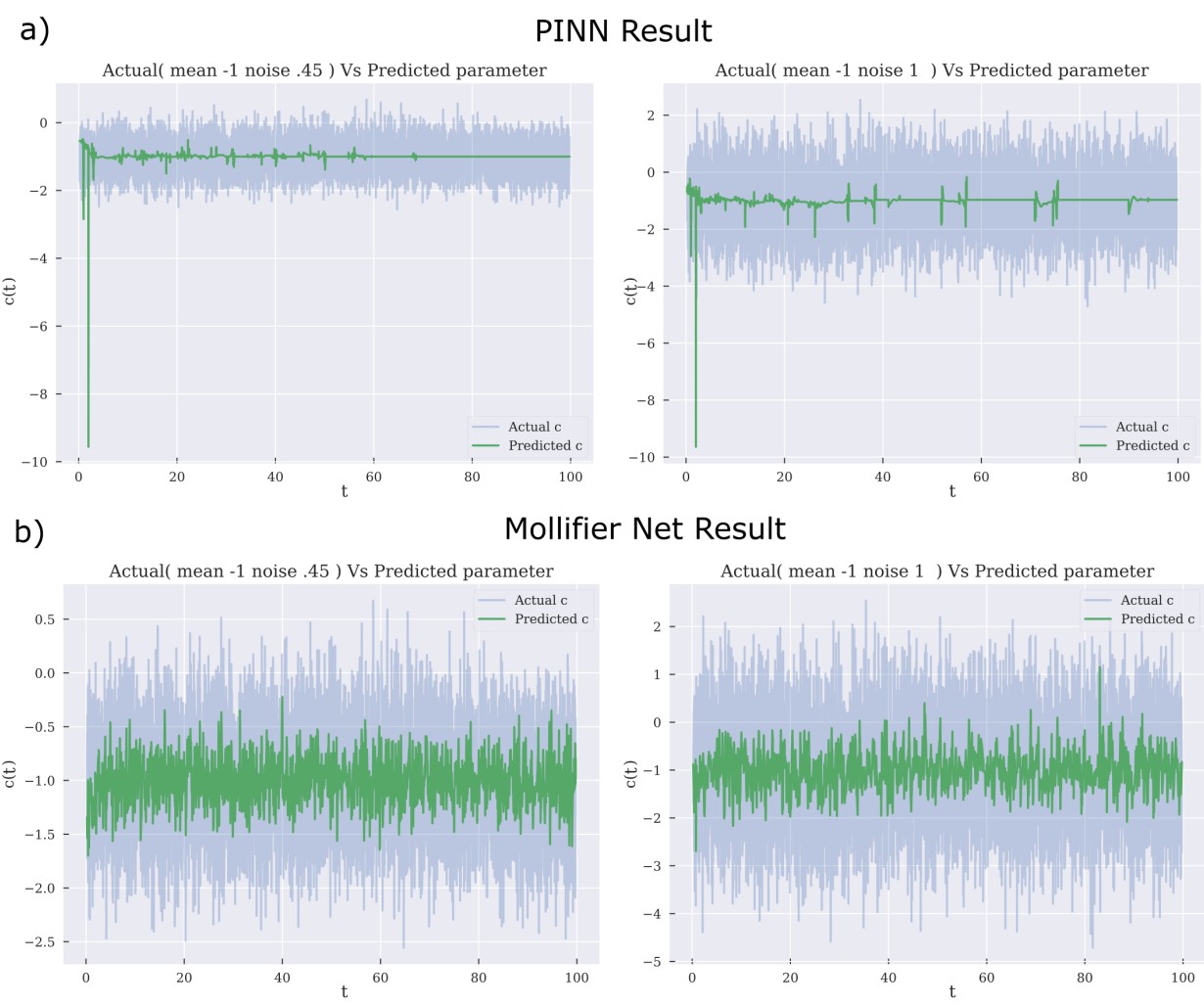

Figure 8: *Noisy forcing term inference in Langevin equation using PINN and Mollified PINN*

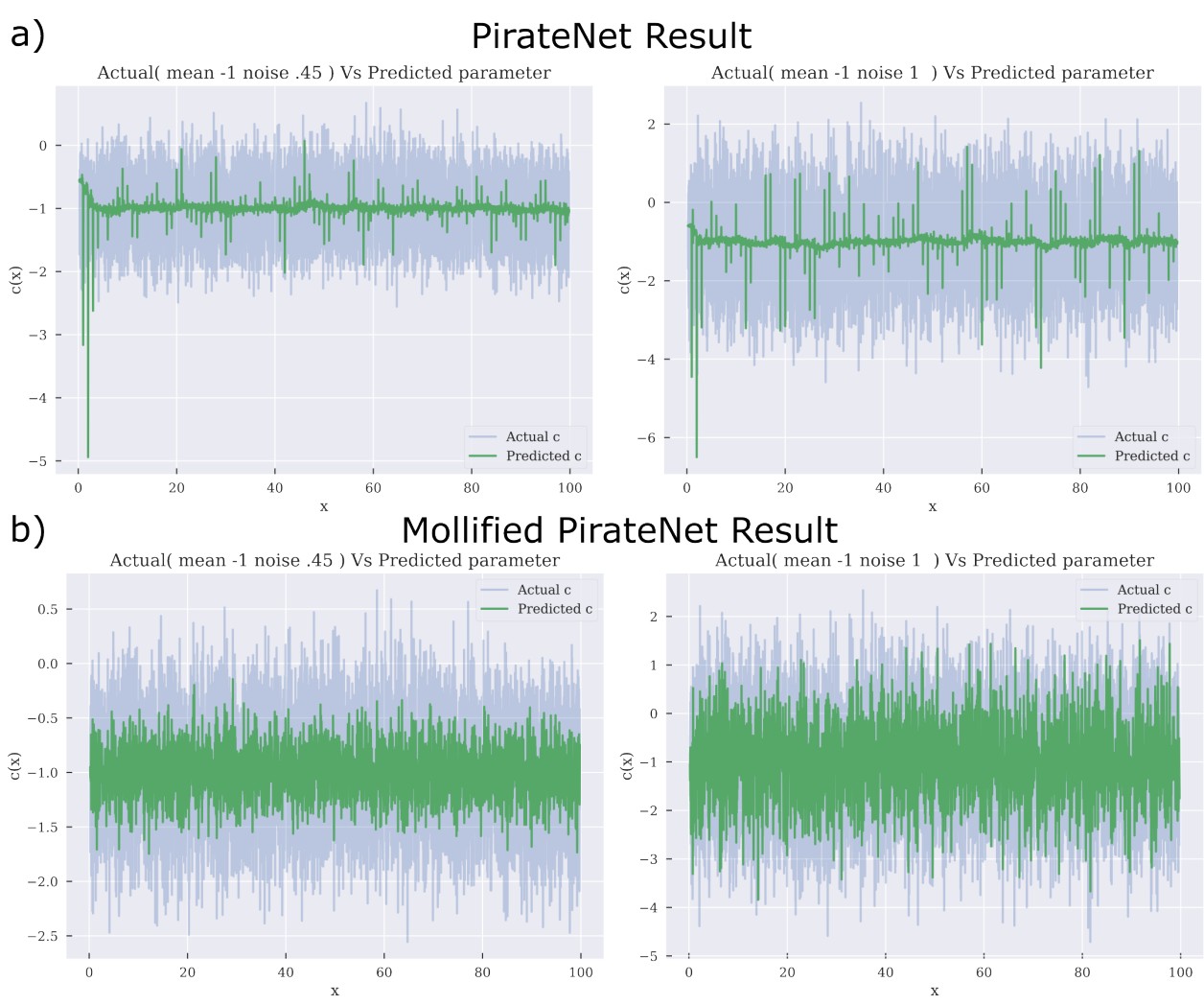

Figure 9: *Noisy forcing term inference in Langevin equation using PirateNet and Mollified PirateNet*

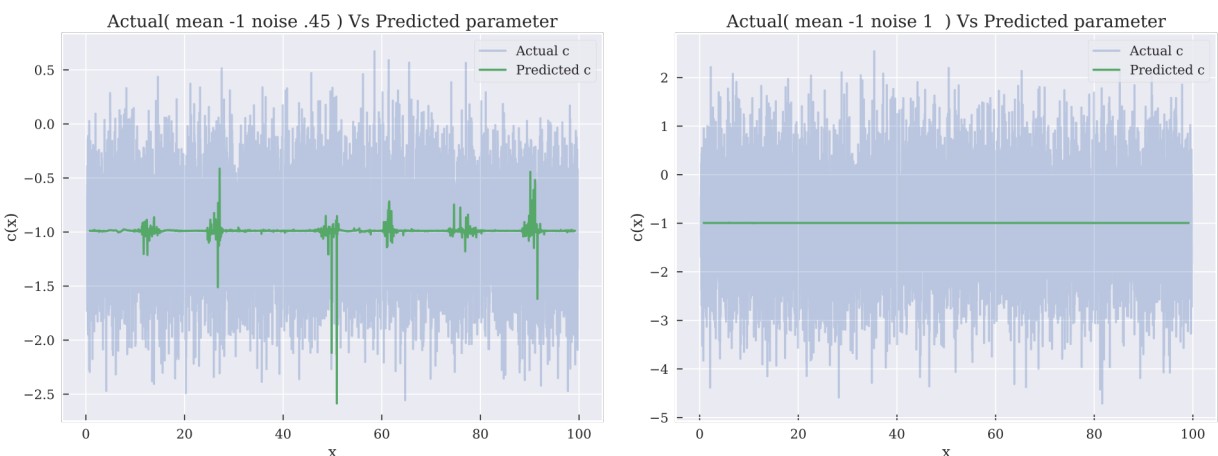

Figure 10: *Noisy forcing term inference in Langevin equation using PINNsFormer*

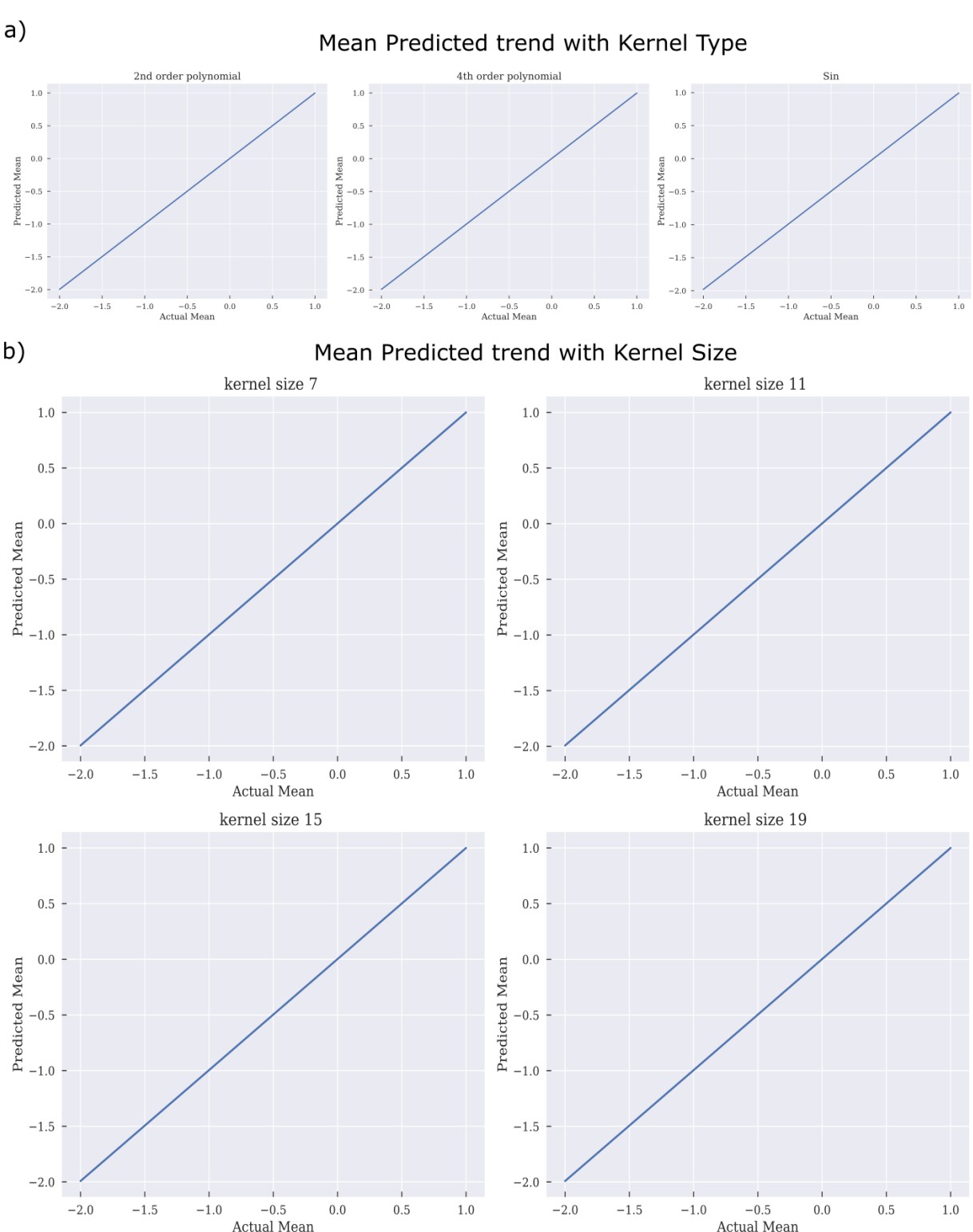

Figure 11: *Mean Forcing term prediction with different kernel choices with Mollified PINNs*

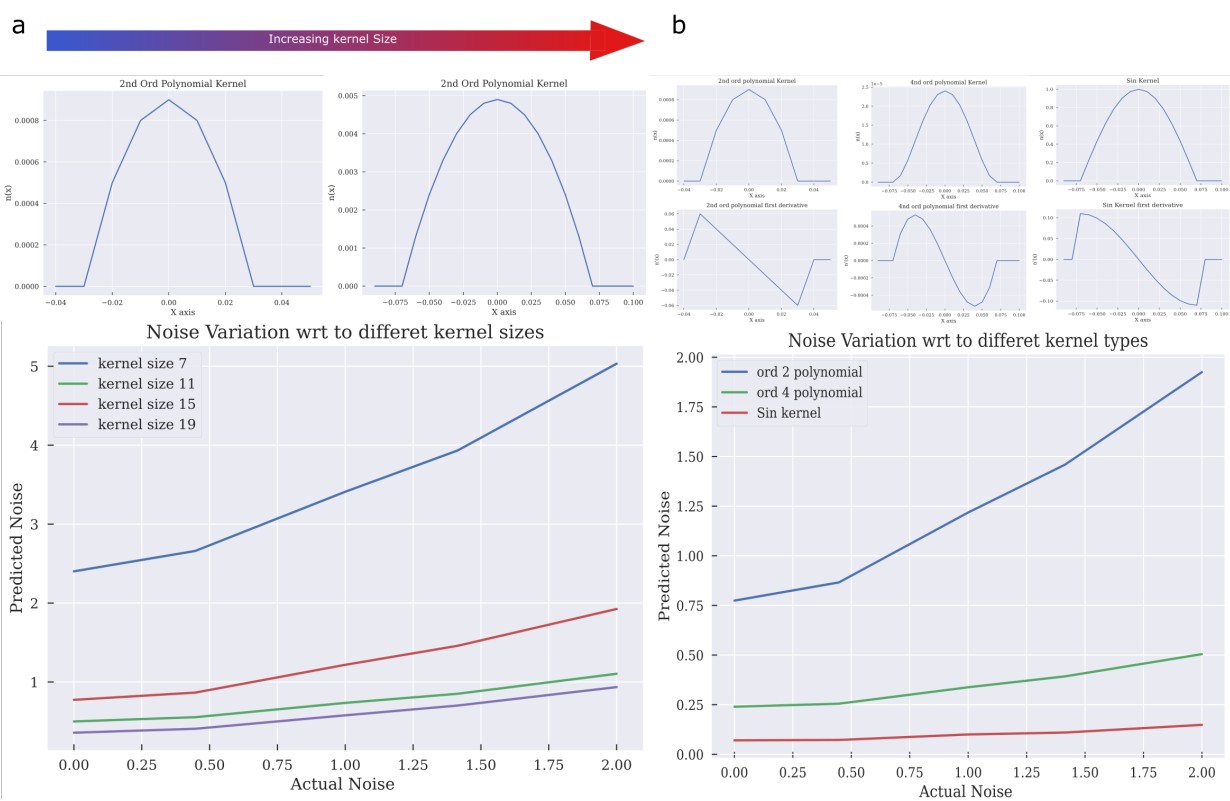

Figure 12: *Estimating the Noise in the forcing term prediction with different kernel choices with Mollified PINNs*

PINN Result on temporal trend

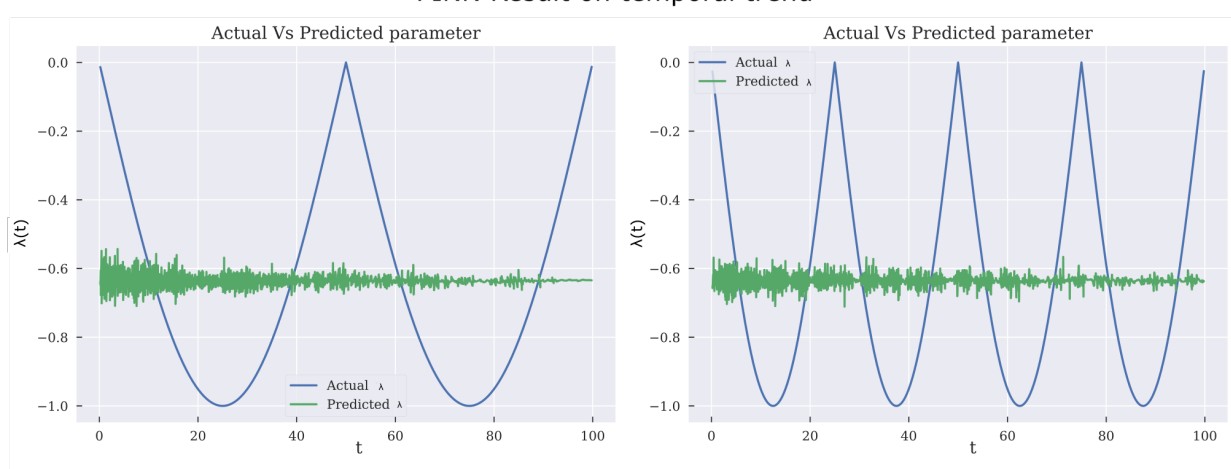

PINN Result on temporal trend with white Noise

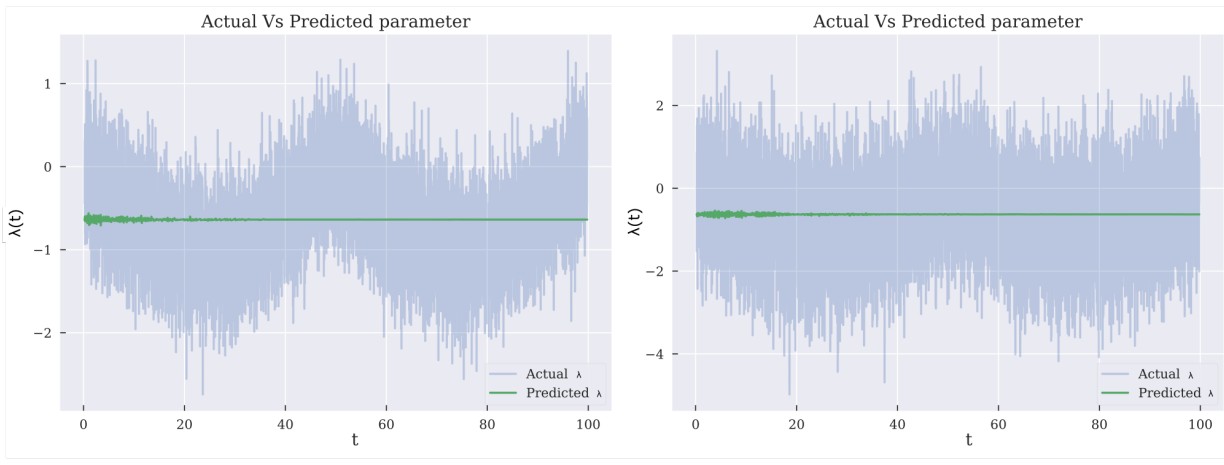

Figure 13: *Estimating with and without noise temporally varying forcing term with PINNs*

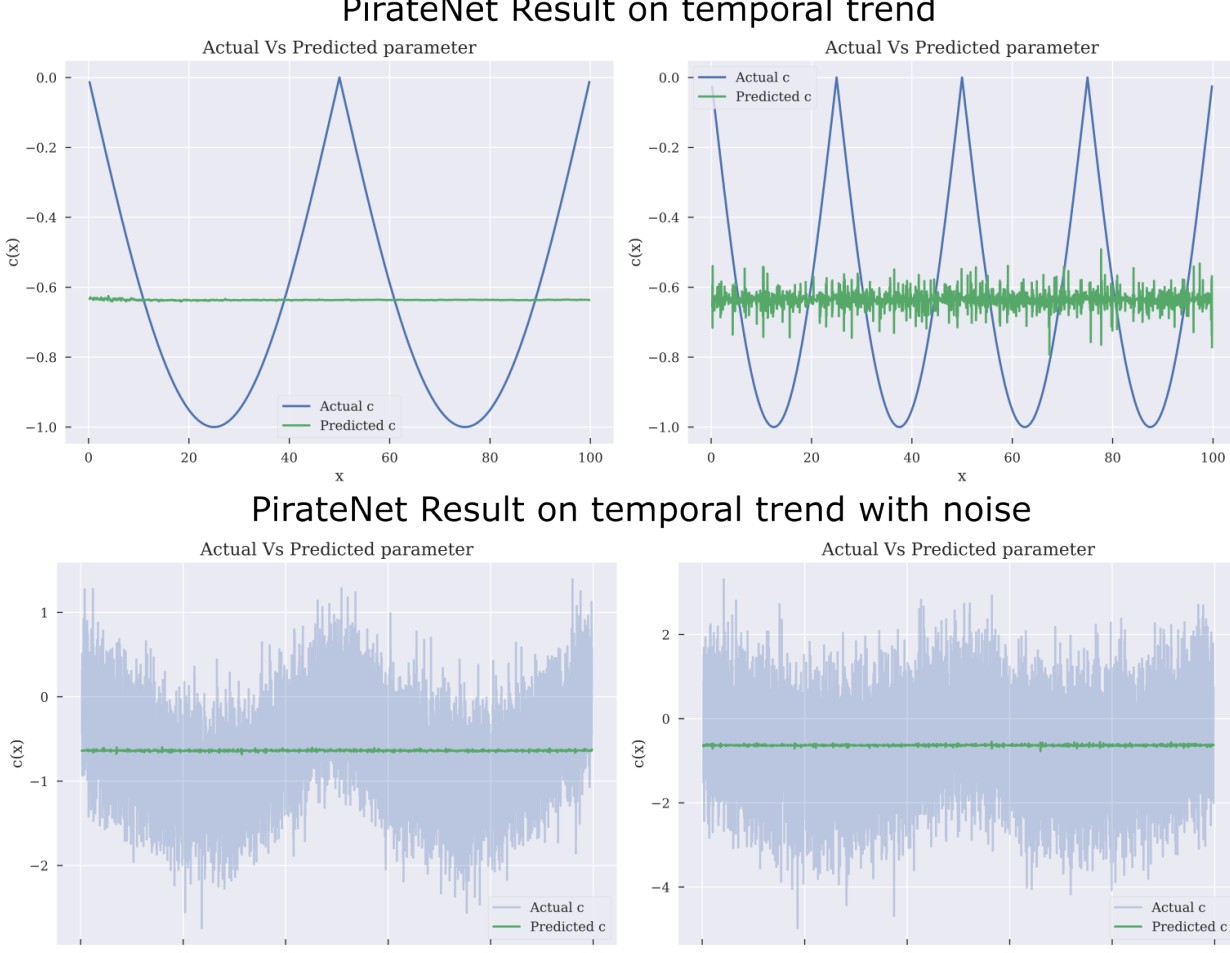

Figure 14: *Estimating with and without noise temporally varying forcing term with PirateNet*

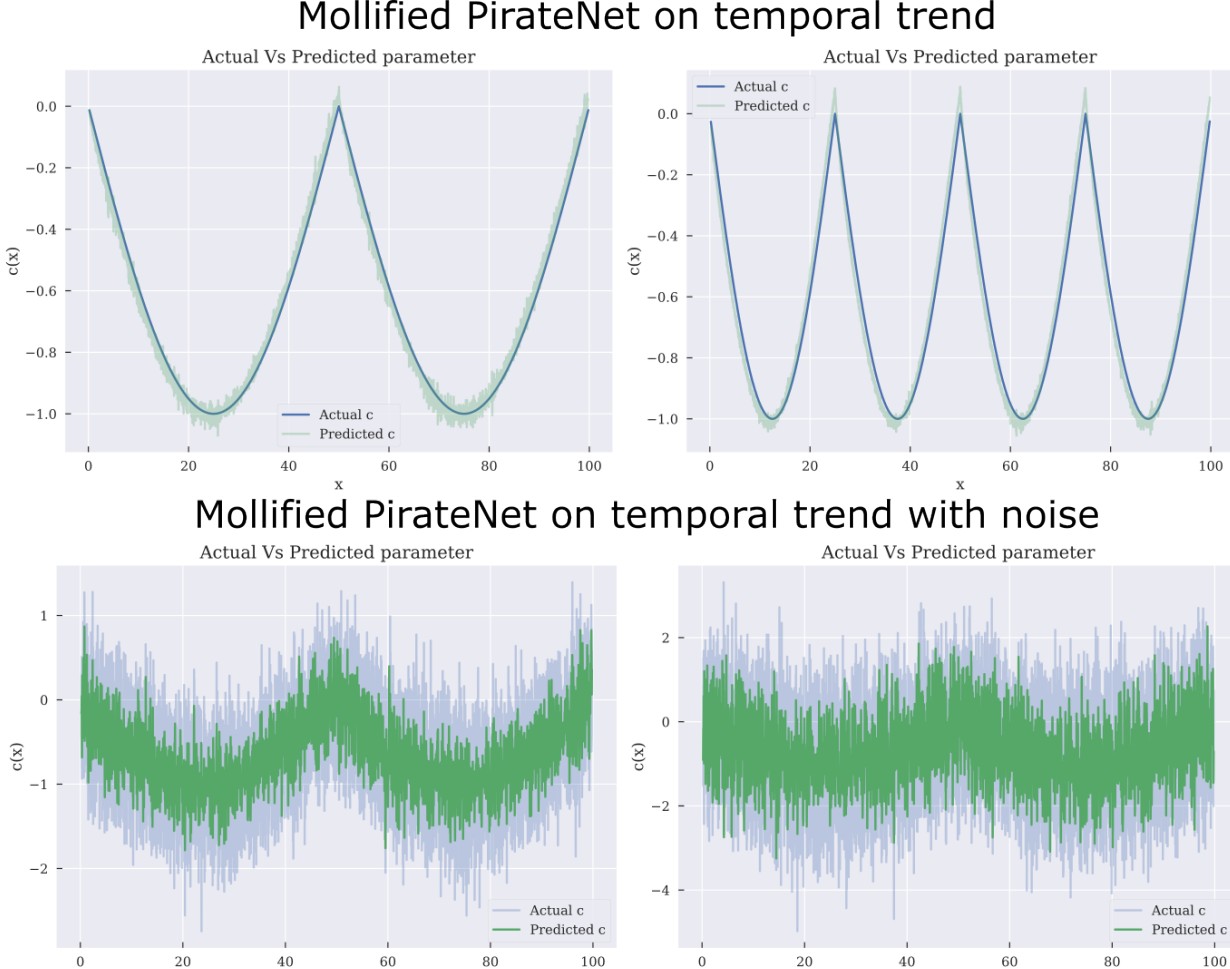

Figure 15: *Estimating with and without noise temporally varying forcing term with Mollified PirateNet*

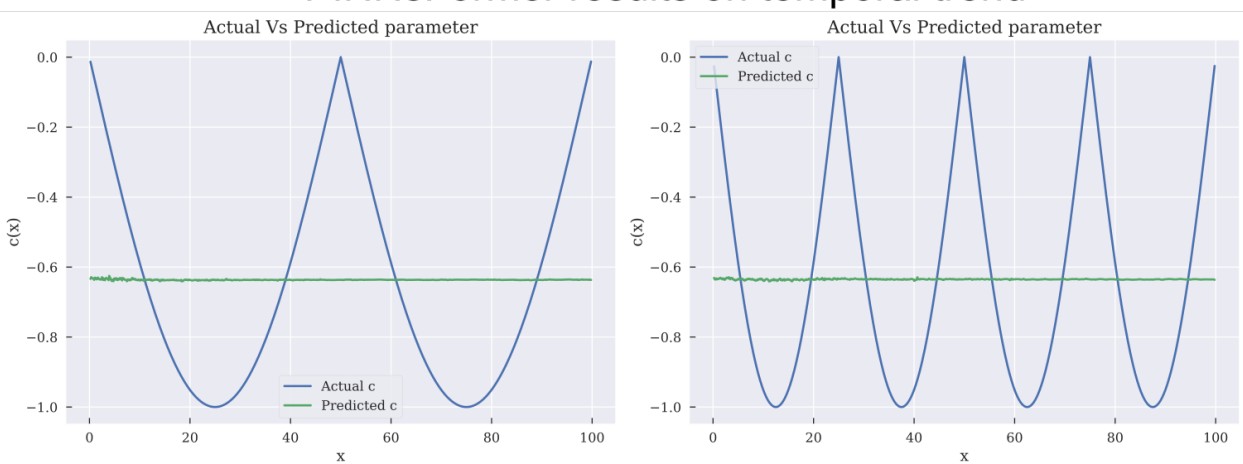

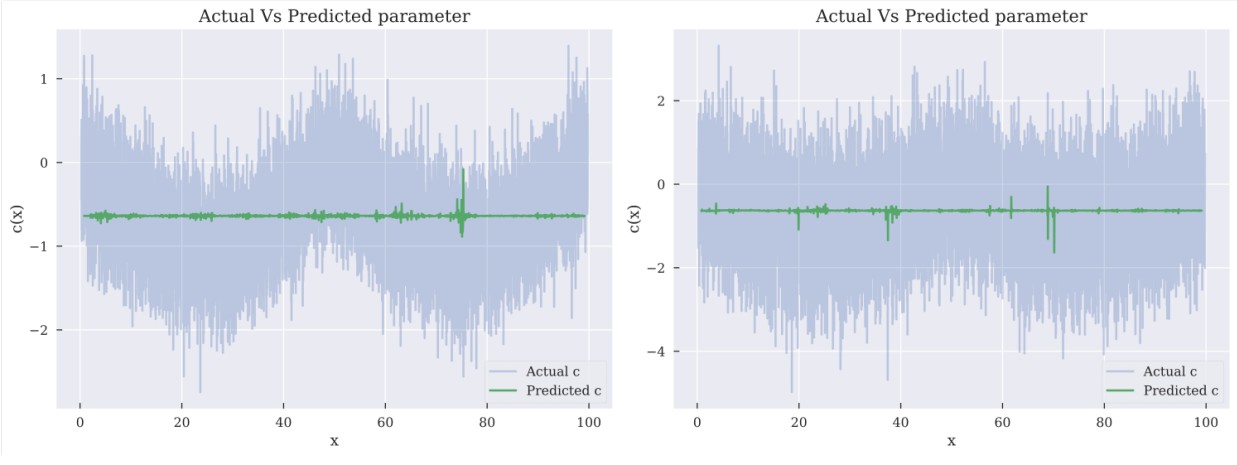

Figure 16: *Estimating with and without noise temporally varying forcing term with PINNsFormer*

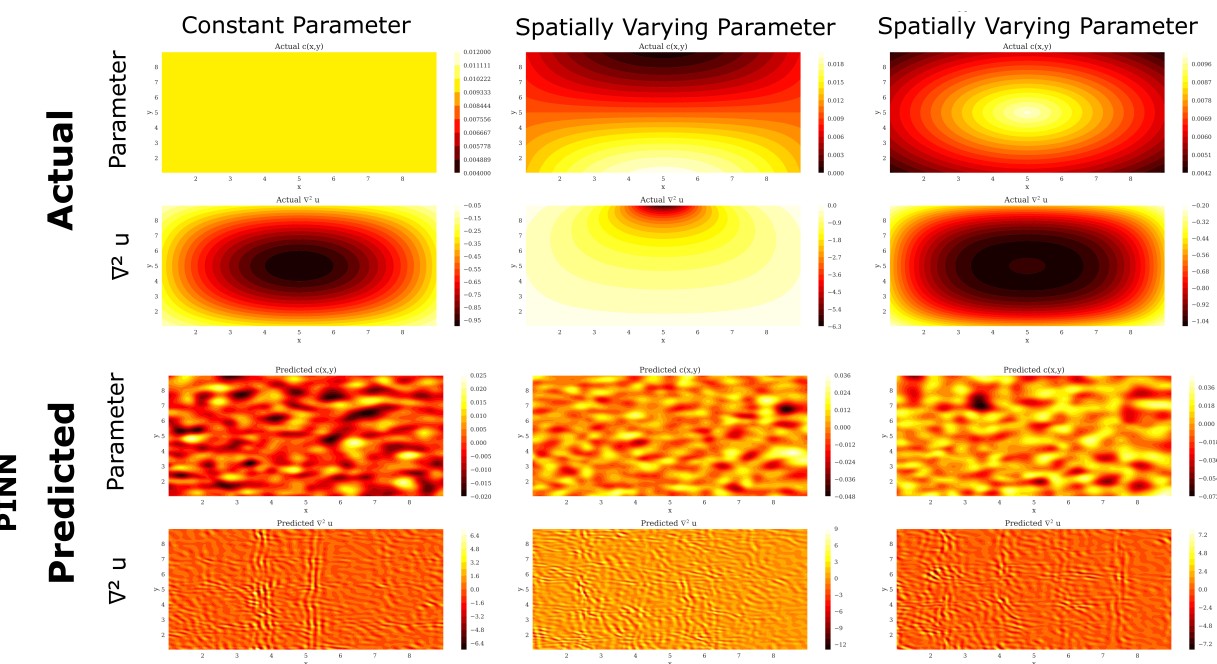

Figure 17: *Estimating constant and spatially varying diffusivity terms and corresponding Laplacians for the heat equation using PINNs*

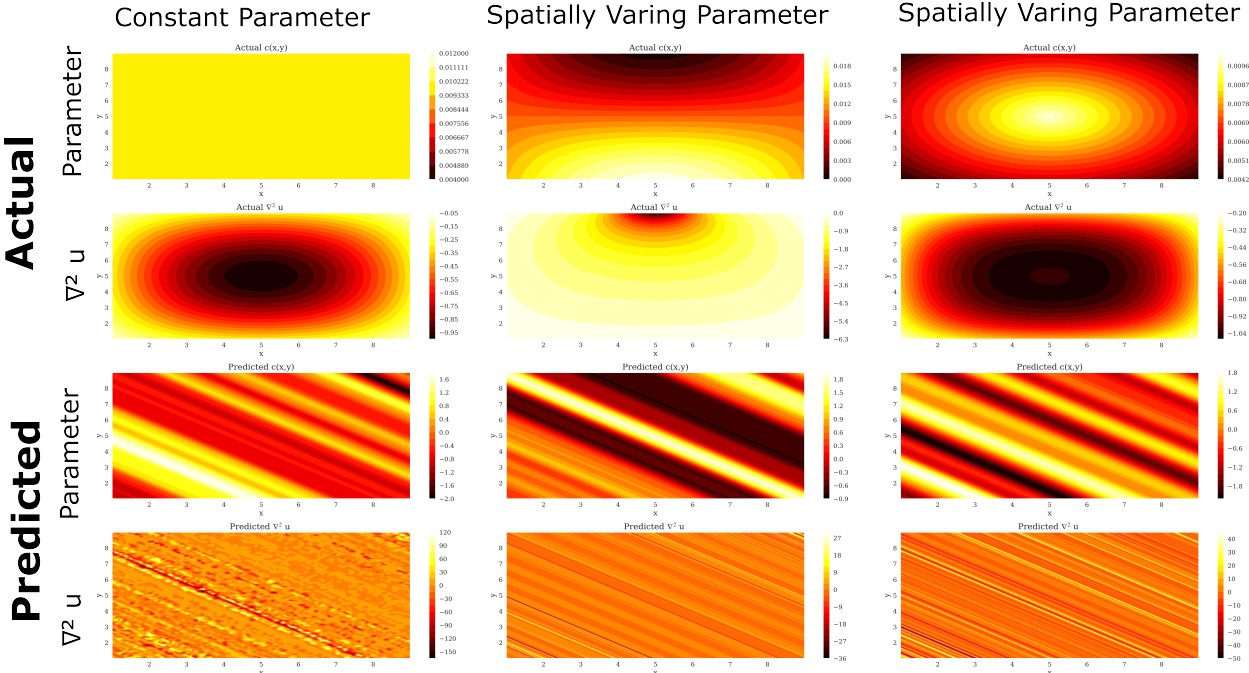

Figure 18: *Estimating constant and spatially varying diffusivity terms and corresponding Laplacians for the heat equation using PirateNets*

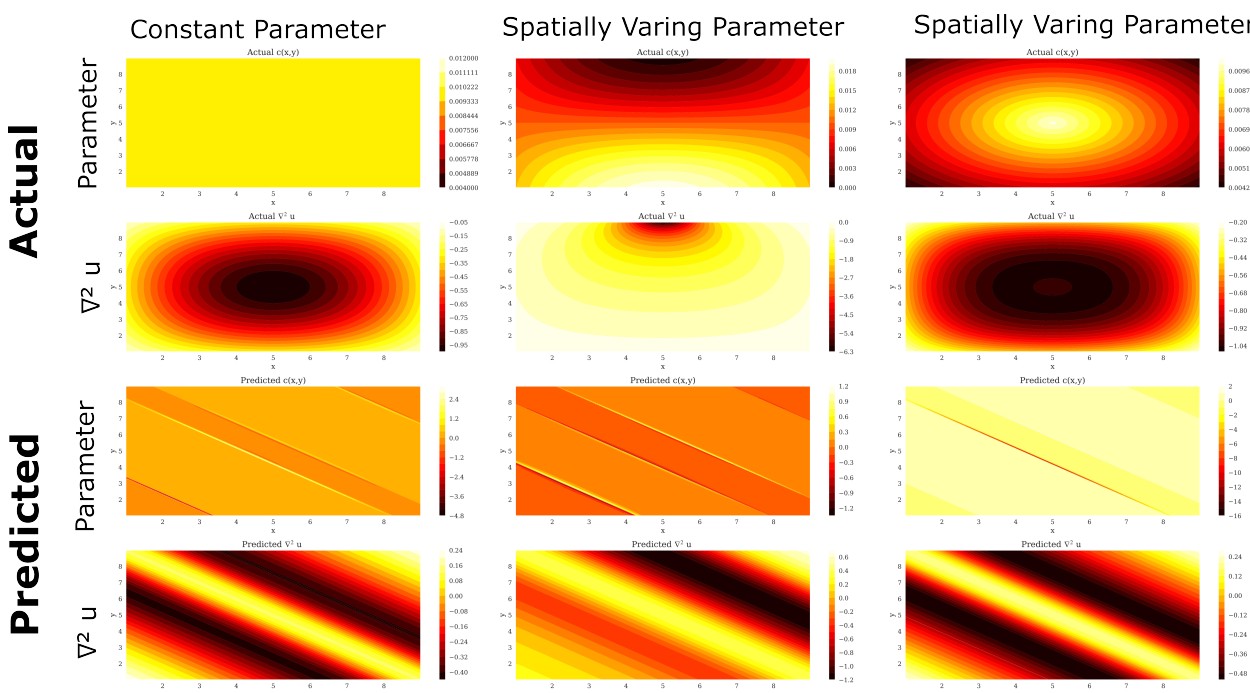

Figure 19: *Estimating constant and spatially varying diffusivity terms and corresponding Laplacians for the heat equation using mollified PirateNets*

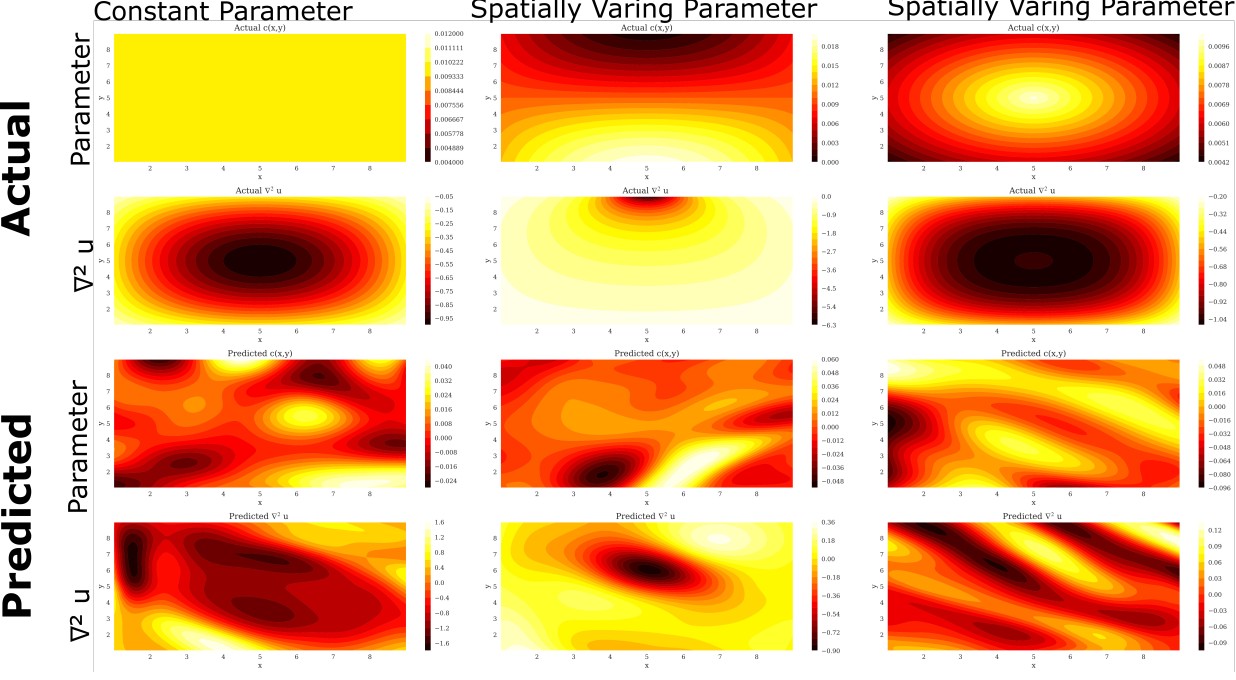

Figure 20: *Estimating constant and spatially varying diffusivity terms and corresponding Laplacians for the heat equation using PINNsFormer*

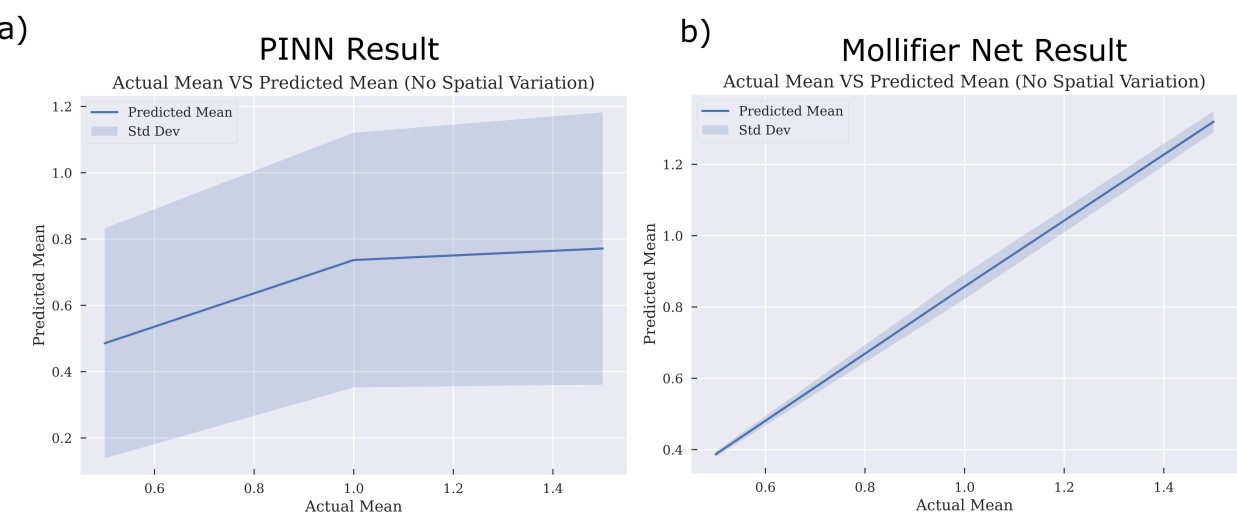

Figure 21: *Predcited vs. Actual mean for the reaction rates in simulations for PINNs and Mollified PINN*

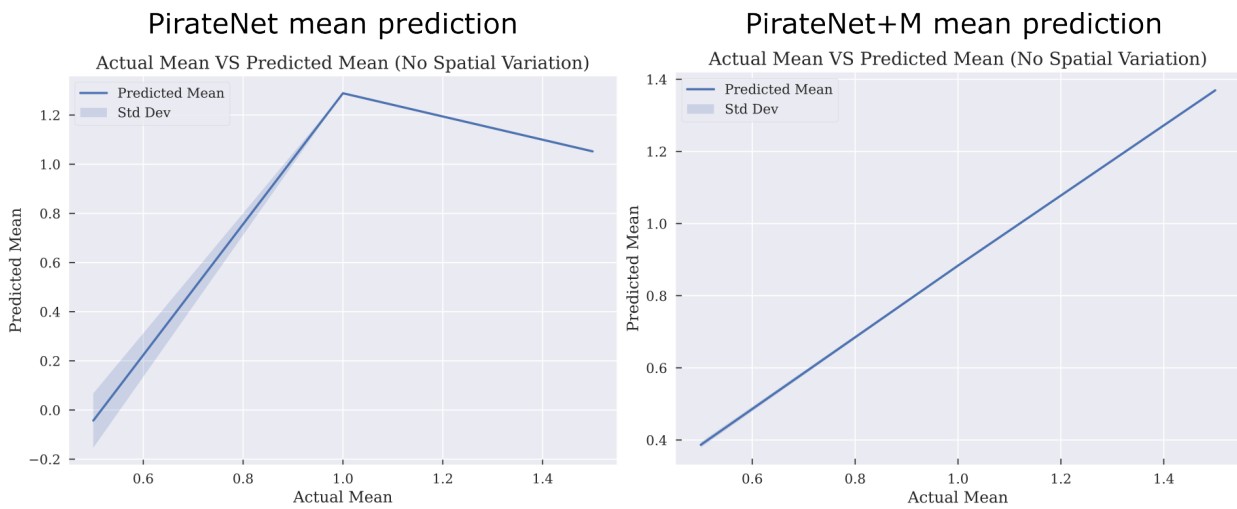

Figure 22: *Predcited vs. Actual mean for the reaction rates in simulations for PirateNet and Mollified PirateNet*

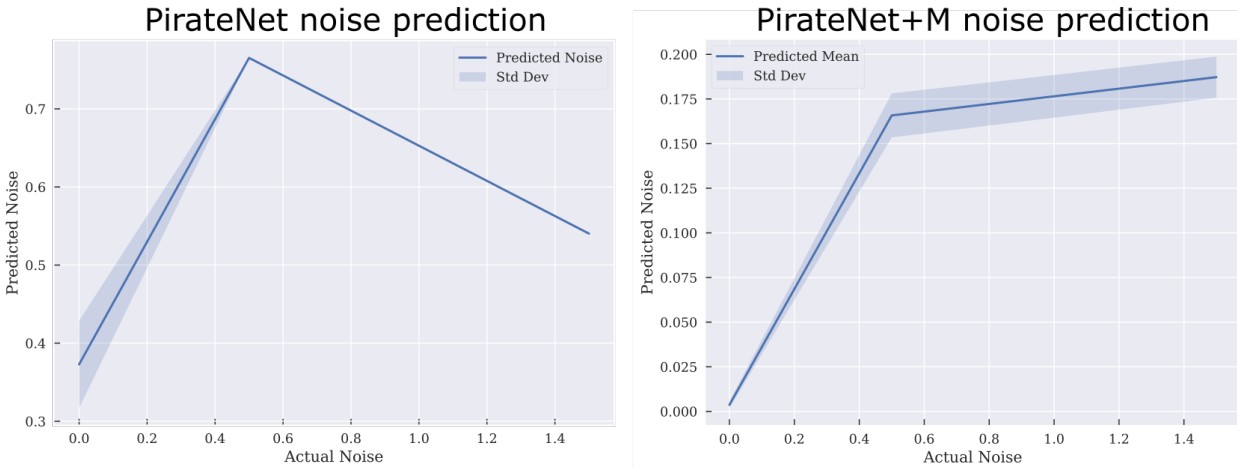

Figure 23: *Predcited vs. Actual noise for the reaction rates in simulations for PirateNet and Mollified PirateNet*

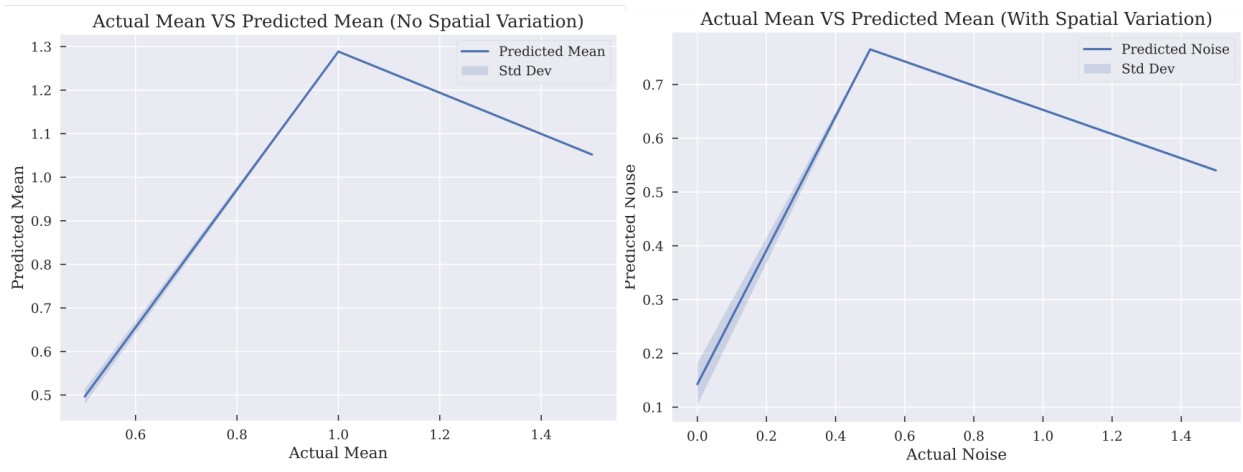

Figure 24: *Predicted vs. Actual mean and noise for the reaction rates in simulations for PINNsFormer*

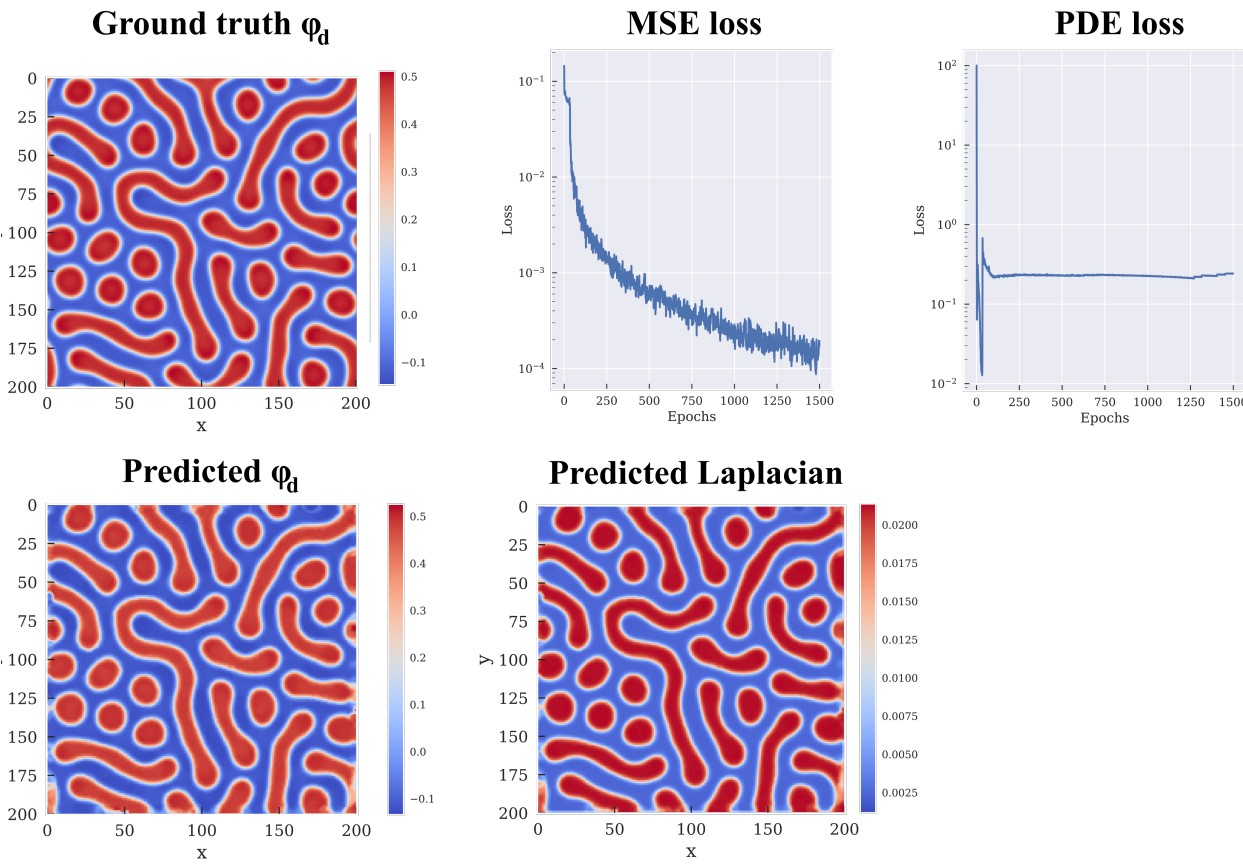

Figure 25: *Mollified PINNs capture the spatial reaction rates for reaction-diffusion equation.*

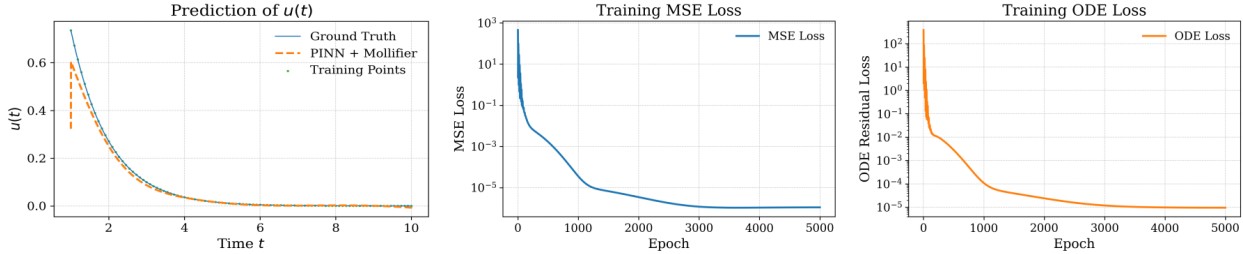

Figure 26: *Using Mollified PINNs for a forward problem.*

