# OpenReview forum: "Mollifier Layers: Enabling Efficient High-Order Derivatives in Inverse PDE Learning"
_TMLR — Accepted by TMLR_

### Review · Reviewer_osLu · 2025-11-28

**Summary Of Contributions:**

This manuscript proposes mollification as a replacement of autodiff for computation of derivatives with respect to time and space in PINN-style models.

It provides evidence that mollification leads to lesser use of memory, faster training time, better mean, temporal and spatial correlation, as well as better Laplacian correlation. It also shows consistent performance for different orders of the underlying problems, unlike (and unsurprisingly) FD.

Although I liked the idea described in the paper, I found sections 3 and 4 (and linked 5) to be problematic.

The key issue with section 3 is incompleteness. After reading that I had only a vague understanding of how to apply mollification in practice. The theoretical part occupies most of the section but largely focuses on nuances of when mollification is applicable. Even then I learnt more about that from reading a dedicated paper on the topic rather than the manuscript. I was hoping that after presenting eq 4, 5 and 6 (with second rather than third derivative), showing some examples of mollifying functions and their analytic derivatives, you would go to describing how these equations are integrated in Figure 1 (implementation through convolution with different kernel sizes).

The key issue with section 4 is again incompleteness. You chose a number of tasks to present but did so in a rather brief fashion redirecting the reader to your appendix on numerous occasions. I have found this to be a very distracting experience. I would have preferred to see one experiment to be fully described in the main body of the paper with others to be placed in the appendix.

**Additional Comments:**

No additional comments.

**Audience:**

Yes

**Audience Explanation:**

As I mentioned above, I liked the idea presented in this manuscript and the results shown also suggest this to be a good approach to capturing higher order derivatives more accurately than what is possible with autodiff.

**Broader Impact Concerns:**

No concerns.

**Claims And Evidence:**

No

**Claims Explanation:**

In my opinion table 2 provides good evidence that mollification works as expected. However, as mentioned above, I believe the manuscript

1) could have been more clear about the technical details behind mollification;
2) could have provided a more comprehensive description of each experiment chosen to go into the main body of the paper.

As such I cannot positively answer to this question.

**Requested Changes:**

1) Improve integration of theory and practice by providing more details how mollification is implemented, what mollifying functions were explored (not in appendix), what their derivatives are, how convolutions were implemented.

2) Focus experimental part on one, maximum two experiments. For instance, the 4th order equation is a good choice. Please describe the experimental setup and experiments performed carefully and clearly.

3) Please provide some technical/mathematical insight into poor performance of autodiff and compare that to mollifying.

4) Switch from 3rd order to 2nd order derivative in eq 6.

5) Please ensure equal ranges of x and y axis on any plot of predicted vs actual noise. I would like to see (non)-diagonality easier without the need to check ranges every single time.

6) Please explain every single symbol used in the main body in the main body (e.g. \Lambda, \sigma on page 7).

7) Please explain a rather weaker performance of mollifying in PDE loss minimisation in Fig 2c including what it implies.

8) Please explain why \phi_n is missing in section 4.3 after its first mention.

9) Please explain/more closely link the text in section 5 to figure 4.

10) Please double check all of your figures in the appendix. For example, Figure 11 shows identical mean prediction performance, which does not make sense.

11) Please add space between text and citation everywhere in the manuscript (e.g. noisy measurements(Kaipio... -> noisy measurements (Kaipio...)

---

> ### Author Response · Authors · 2026-01-20
> **Point-by-point response to Reviewer osLu**
>
> We thank the reviewer for the careful reading and constructive feedback. We appreciate the positive assessment of the core idea, using mollification as an alternative to autodiff for high-order derivatives, and the note that the empirical evidence (Table 2) is compelling.
>
> 1.  **Sec. 3 implementability (mollifiers, derivatives, kernels/convolution).**
>
> In the revised manuscript, Sec. 3 has been rewritten to be implementable from the main text. We (i) added a concise implementation recipe showing how order-$k$ derivatives are computed via fixed-kernel convolutions, (ii) provided an explicit mollifier choice and a clear description of how the corresponding derivative kernels are obtained, (iii) clarified how the smoothing scale/bandwidth determines kernel size, and (iv) added a short mapping connecting Eqs. (4-6) to the pipeline in Fig. 1d.
>
> 2. **Reliance on appendix.**
>
> We agree that deferring details to the appendix could make the presentation feel incomplete. We retained the breadth of experiments (first-, second-, and fourth-order PDEs) as showing applicability across orders is at the core of our contribution but made the main text more self-contained by adding compact “Setup” descriptions for each experiment (PDE/unknowns, observations/noise model, objective, metrics). Appendix references are mainly for auxiliary plots/ablations and extended tables rather than essential experimental specification.
>
> 3. **Autodiff degrades at high order; contrast with mollification.**
>
> We thank the reviewer for this suggestion. The manuscript provides both technical motivation and empirical evidence for why recursive autodiff can fail in high-order/noisy inverse PDE settings, highlighting superlinear compute from repeated backpropagation, instability/inaccuracy of high-order derivatives, and memory blow-up from storing gradient graphs (illustrated in Fig. 1 and quantified in Table 1 and related results). In contrast, Sec 3 shows that Mollifier Layers replace recursive differentiation with convolution against analytic derivatives of smooth, compactly supported kernels, acting as an implicit low-pass filter that stabilizes derivative estimates under noise. Appendix A.8 further provides a convergence/error analysis.
>
> 4. **Eq. (6) consistency across Eqs. (4-6).**
>
> Thank you for noting the confusion. In the revision, we corrected the derivative order in Eq. (6), revised Eqs. (4-6) to use consistent notation, and added a clarifying sentence that the method applies to arbitrary derivative orders via the corresponding derivative-kernel construction.
>
> 5. **Predicted-vs-actual noise plots.**
>
> Thank you for this suggestion. We would like to clarify that the predicted-versus-actual noise scatter plots are intended to illustrate correlation and trend agreement, rather than pointwise equality or absolute calibration. Inverse recovery under noisy observations is inherently ill-posed and non-unique, so multiple latent noise realizations can be consistent with the same data and governing equations; consequently, exact diagonal alignment is neither expected nor required. Our evaluation therefore focuses on whether the method captures the correct dependence and variability structure (e.g., tracking higher vs. lower noise levels), which is quantified by the correlation metrics reported in Table 2. The axis ranges in these plots are chosen to keep the scales comparable and to make the correlation structure visually clear; accordingly, these figures should be interpreted primarily through the strength of the correlation they depict, as qualitative support for the quantitative results.
>
> 6. **Missing symbol definitions (e.g., $\Lambda$, $\sigma$).**
>
> We have ensured symbols are defined at first use in the main text.
>
> 7. **Fig. 2c: weaker PDE-loss minimization and implications.**
>
> Inverse PDE problems are often ill-posed/non-unique, so minimizing PDE residual alone does not uniquely identify the desired inverse solution. Fig. 2c is included to show training dynamics; our primary evaluation uses parameter/prediction error and robustness metrics, reported in Table 2.
>
> 8. **Why $\phi_n$ disappears in Sec. 4.3.**
>
> We clarified that $(\phi_h,\phi_e,\phi_n)$ satisfy $\phi_h+\phi_e+\phi_n=1$, and that Sec. 4.3 uses the reduced order parameter $\phi_d=\phi_h-\phi_e$; $\phi_n$ is implicit and recoverable from the constraint.
>
> 9. **Linkage between Sec. 5 and Fig. 4.**
>
> We revised Sec. 5 to explicitly reference Fig. 4 panels and added brief guidance on what each panel demonstrates.
>
> 10. **Appendix Fig. 11 verification.**
>
> We confirm Fig. 11 is correct; overlapping mean-prediction curves are expected because mean prediction saturates for both methods, while differences emerge in higher-order metrics.
>
> 11. **Typos/formatting (citation spacing, etc.).**
>
> We performed a proofreading/formatting pass to correct spacing around citations.
>
> We thank the reviewer again for the thoughtful feedback, which helped improve the clarity and presentation of the manuscript.

---

### Review · Reviewer_1F51 · 2025-12-12

**Summary Of Contributions:**

**Summary**
The paper introduces a mollifier layer, an architecture-agnostic extension to the standard network used in physics-informed machine learning (PhiML). The mollifier layer serves as a mechanism to compute efficient and noise-robust higher-order derivatives and serves as a replacement for the standard autodiff algorithm (which computes the derivatives but is both compute and memory inefficient and lacks robustness to noise).

**Baselines**
Sufficient baselines, showcasing the agnostic nature of the proposed method. Three PhiML architectures were reported. For each architecture, a comparison was made between the standard architecture (i.e., with autodiff) and its mollifier-augmented variant.

**Reproducibility**
The source code for the approach was published, which supports the reproducibility of the results and further research in the area.

**Limitations**
1. The approach may underperform in computing the [higher order] derivatives when the data contains high frequency variation due to smoothness bias induced via the mollifier layer. As a mitigation strategy, the paper introduces an alternative estimation mechanism that is only applicable when $\lambda$ is separable in the PDE.
2. The approach currently requires manual fine-tuning to obtain the correct kernel size, and the choice of kernel is also very important. The approach is very sensitive to the frequency distribution of the underlying function.

**Additional Comments:**

N/A

**Audience:**

Yes

**Audience Explanation:**

The paper addresses a relevant problem in the sub-field of Physics Informed Machine Learning (PhiML), a growing sub-area of machine learning research. Furthermore, there are PhiML researchers within the TMLR community, as evidenced by previously accepted publications in the research area.

**Claims And Evidence:**

Yes

**Claims Explanation:**

The claims were validated by the empirical results across three off-the-shelf PhiML baselines. For each baseline, the mollified-augmented version showed better robustness to noise than the standard baseline (without the mollifier layer).

**Requested Changes:**

1. Typographical error in page 4, paragraph 2. “Here, $N_u$ are the training data points available for the actual solution $u(t, x)$ and $N_f$ *all* all the points in”. The first “all” should be “are”

---

> ### Author Response · Authors · 2026-01-20
> **Point-by-point response to Reviewer 1F51**
>
> We thank the reviewer for their supportive assessment of the contribution, the adequacy of baselines, and the emphasis on reproducibility.
>
> 1. **Typo**
>
> We have corrected the p.4 (para 2) sentence to “are all…” and removed the redundant “all all …”.
>
> 2. **High-frequency limitation**
>
> We agree mollification induces a low-pass (smoothness) bias and may underperform when the underlying solution contains substantial high-frequency content. Our focus is the inverse-PDE regime with noisy observations where high-order derivatives are variance-dominated; mollification provides a controllable bias-variance tradeoff that improves robustness under noise. We have clarified this in the limitation.
>
> 3. **Separable-parameter estimator**
>
> We agree this optional mechanism applies to a subset of problems; we have clarified this in the text and have mentioned that it as a special case and emphasize that the core mollifier-layer method remains architecture-agnostic and does not rely on separability.
>
> 4. **Kernel tuning**
>
> We agree that kernel choice matters. We provide a kernel-size sensitivity analysis in Appendices A.5 and B.1.3 and summarize the resulting observations.
>
> We thank the reviewer again for the thoughtful feedback, which helped improve the clarity and presentation of the manuscript.

---

### Review · Reviewer_vw5y · 2026-01-13

**Summary Of Contributions:**

The paper address a common scientific challenge related to parameter estimation in inverse problems involving partial differential equations where unknown model parameters may vary in space or time and is to be estimated given sparsely sampled data.

The authors proposes to deal with high-order derivatives in physics-informed machine learning approaches with mollifier layers instead of relying on automatic differentiation with the aim to reduce the overall computational costs and improve the performance through more accurate estimation of derivative computations. By introducing mollifier layers it is possible to reframe the derivative computation as a smoothening integration that may be done by attaching mollifier layers to the output layer without any other architectural modifications. Comparisons of this proposed network design technique is done against a  physics-informed neural network (PINN) approach for different inverse PDE problems that are representative for commonly encountered problems in science and including a real-world application of relevance in biomedical areas.

The new mollifier layer technique is aimed at lowering the computational and memory costs of dealing with high-order derivatives and also give support to improved accuracy for modeling spate-temporal varying systems and systems with noisy parameters. The proposed technique is interesting as it is a new/alternative way to use and possibly improve physics-informed neural networks for a class of problems.

**Additional Comments:**

Overall well-written paper and interesting and relevant use of mollifiers with physics-informed machine learning.

**Audience:**

Yes

**Audience Explanation:**

Yes the TMLR serve a technical community and the proposed mollifier technique is an interesting new idea that warrants evaluation.

**Broader Impact Concerns:**

No suggestion.

**Claims And Evidence:**

Yes

**Claims Explanation:**

Physics-informed machine learning techniques have seen a revival in recent years and given strong attention in the research communities. The proposed mollifier techniques is aimed at improving the naive PINN approach and with emphasis on parameters estimation done when solve inverse PDEs, which constitute an important set of scientific problems that span science and engineering.

The mollifier techniques are well-known are important, and it is interesting to see it being used to improve PINNs for parameter estimation when solving inverse PDEs. This is a new idea to the best of my knowledge and for this reason the study is highly relevant.

The benchmark PDE problems considered, highlights that mollifier layers can improve PINN-type approaches (cf. their Table 2), and in particular when performing parameters estimation with noisy data as a result of the smoothening property inherited from the mollifiers. In particular it is demonstrated that mollifier layers improve techniques such as PINN (with techniques laid out in section A.4) and PirateNet, and is better than the finite-difference based PINN approach where finite difference are sensitive to noisy data and hence perform poorly. The problems considered incudes arguably non-trivial time-varying parameters and spatially varying parameters, and problems of PDEs. How would the model work for problems with multiple parameters that is to determined simultaneously? ... this is not considered.

A short coming of the paper is the lack of mentioning of other (competitive) techniques used for parameter estimation that is not based on neural networks and possibly including a comparison with such techniques. For example, there is the classical newton-based techniques such as framed in ODIL [1], e.g. see references given here. The authors of [1] challenges current PINN paradigm(!). [2] proposed recently using the closed-form least square solution procedure for ODEs (main emphasis) and PDEs (one example) and compared to (naive) PINNs and outperform also PINN in parameter estimation.

[1] Solving inverse problems in physics by optimizing a discrete loss: Fast and accurate learning without neural networks
Petr Karnakov , Sergey Litvinov , Petros Koumoutsakos Author Notes
PNAS Nexus, Volume 3, Issue 1, January 2024, pgae005, https://doi.org/10.1093/pnasnexus/pgae005
URL: https://academic.oup.com/pnasnexus/article/3/1/pgae005/7516080

[2] Physics-Informed Regression: Parameter Estimation in Parameter-Linear Nonlinear Dynamic Models
Jonas Søeborg Nielsen, Marcus Galea Jacobsen, Albert Brincker Olson, Mads Peter Sørensen, Allan Peter Engsig-Karup, arXiv, 2025.
https://arxiv.org/abs/2508.19249

**Requested Changes:**

It is suggested to expand the literature review and mention other state-of-the-art approached, e.g. ODIL, and discuss the relation to this work. See other comments given about shortcomings.

It is suggested to compare to non-neural network based techniques in the benchmarks.

---

> ### Author Response · Authors · 2026-01-20
> **Point-by-point response to Reviewer vw5y**
>
> Thank you for the thoughtful and supportive review. We are glad the reviewer finds mollifier layers useful for stabilizing high-order derivatives, particularly in noisy inverse PDE settings. Below is a point-by-point response to the suggestions from the reviewer:
>
> 1. **Related work: ODIL / non-NN inverse methods**
>
> We agree that the current related-work discussion underrepresents non-neural approaches. In the revision we have added text contrasting mollifier layers with discrete optimization / learning-without-neural-network approaches such as ODIL (Karnakov et al., 2024) to the related works section. We additionally note that our contribution is orthogonal: rather than proposing a new solver family, we provide a drop-in differentiable derivative operator that can be attached to existing NN-based inverse PDE pipelines to avoid recursive autodiff and improve robustness to noise.
>
> 2. **Additional baselines**
>
> We agree that comparisons to alternative inverse-PDE paradigms such as ODIL can be informative. That said, ODIL-type methods and our approach address a different design choice than the one we study: ODIL changes the representation/solver class (direct discretization + optimization), whereas Mollifier Layers are a drop-in derivative operator intended to improve gradient stability and high-order derivative estimation within neural inverse-PDE pipelines (e.g., PINNs and neural-operator-style parameterizations) without changing the underlying model class. Because of this mismatch, a direct head-to-head benchmark can be misleading unless one carefully controls for discretization, parameterization, and optimization settings.
>
> To address the reviewer’s point constructively, in the revised manuscript we added a dedicated discussion of ODIL and closely related physics-informed regression approaches, clarifying (i) when ODIL-style discretization-based solvers may be preferable (e.g., when discretization is trusted and a neural approximation is unnecessary) and (ii) when Mollifier Layers are beneficial (e.g., neural parameterizations requiring stable high-order derivatives, noisy data, or stiff operators). We believe this contextualizes the trade-offs and helps position Mollifier Layers relative to solver-class alternatives.
>
>
> 3. **Multiple parameters**
>
> We agree that multi-parameter inference is important. Our mollifier-layer formulation extends directly to multi-parameter settings (e.g., vector-valued or multi-field unknowns) by applying the same mollified derivative operators to the relevant network outputs and optimizing a joint inverse objective. For completeness, we have included an appendix section (Appendix D) that details this extension and provides an illustrative multi-parameter formulation.
>
> We thank the reviewer again for the thoughtful feedback, which helped improve the clarity and presentation of the manuscript.

---

### Author Response · Authors · 2026-01-20
**Summary of Revisions and Responses to Reviewer**

Dear Action Editor and Reviewers,

Thank you to all reviewers for the careful reading and constructive feedback. We have revised the manuscript to make the method operationally clear from the main text, improve presentation/notation, and better position the contribution relative to existing paradigms. In the revised manuscript, all changes are highlighted in **blue** for ease of inspection.

**Summary of key changes:**

1.	***Implementation clarity (Sec. 3):*** Rewrote Section 3 to be directly implementable from the main paper, including an explicit implementation recipe (fixed-kernel convolutions for derivatives), an explicit mollifier choice, and a clear mapping from eqs to the pipeline in Fig 1d.
2.	***Reduced reliance on appendix:*** Expanded main-text experiment descriptions so each setting can be understood without the appendix; appendix now primarily concerns auxiliary plots/ablations and extended tables.
3.	***Equation/notation fixes:*** Corrected the derivative order/consistency issue in Eq. (6) and clarified the “arbitrary-order” intent; added missing symbol definitions at first use (e.g., ($\Lambda$, $\sigma$)); clarified why ($\phi_n$) is implicit after reduction in Sec. 4.3 via the space-filling constraint.
4.	***Positioning vs non-NN inverse methods:*** Added a dedicated discussion contrasting Mollifier Layers with ODIL-style discretization/optimization approaches and related physics-informed regression methods.
5.	***Additional clarifications requested by reviewers:***  Strengthened the narrative linkage between Sec. 5 and Fig. 4, verified appendix plotting artifacts (e.g., Fig. 11), and performed a proofreading/formatting pass.

We thank the reviewers and the editor again for their time and the actionable suggestions, which improved the clarity and positioning of the work.